**Effects of spatial and temporal variability in surface water inputs on streamflow generation and cessation in the rain-**
**snow transition zone**
**Leonie Kiewiet[1,2], Ernesto Trujillo [3,4], Andrew Hedrick [4], Scott Havens[4], Katherine Hale[5,6], Mark Seyfried [4], Stephanie**
**Kampf[1], Sarah E. Godsey [2]**
*Correspondence to*: Leonie Kiewiet (leoniekiewiet@gmail.com)
[1] Department of Ecosystem Science and Sustainability, Colorado State University, Fort Collins, CO, USA
[2] Department of Geosciences, Idaho State University, Pocatello, ID, USA
[3] Department of Geosciences, Boise State University, Boise, ID, USA
[4] USDA Agricultural Research Service, Boise, ID, USA
[5] Department of Geography, University of Colorado, Boulder, CO, USA
[6] Institute of Arctic and Alpine Research, University of Colorado, Boulder, CO, USA
**Abstract**
Climate change affects precipitation phase, which can propagate into changes in streamflow timing and magnitude. This study
examines how the spatial and temporal distribution of rainfall and snowmelt affect discharge in rain-snow transition zones.
These zones experience large year-to-year variations in precipitation phase, cover a significant area of mountain catchments
globally, and might extend to higher elevations under future climate change. We used observations from eleven weather
stations and snow depths measured from one aerial lidar survey to force a spatially distributed snowpack model
(iSnobal/Automated Water Supply Model) in a semi-arid, 1.8 km$^2$ headwater catchment. We focused on surface water inputs
(SWI; the summation of rainfall and snowmelt on the soil) for four years with contrasting climatological conditions (wet, dry,
rainy and snowy) and compared simulated SWI to measured discharge. A strong spatial agreement between snow depth from
the lidar survey and model (r$^2$: 0.88) was observed, with a median Nash-Sutcliffe Efficiency (NSE) of 0.65 for simulated and
measured snow depths at snow depth stations for all modelled years (0.75 for normalized snow depths). The spatial pattern of
SWI was consistent between the four years, with north-facing slopes producing 1.09 to 1.25 times more SWI than south-facing
slopes, and snow drifts producing up to six times more SWI than the catchment average. Annual discharge in the catchment
was not significantly correlated with the fraction of precipitation falling as snow, but instead with the magnitude of
precipitation and spring snow and rain. Stream cessation depended on total and spring precipitation, as well as on the melt-out
date of the snow drifts. These results highlight the importance of the heterogeneity of SWI at the rain-snow transition zone for
streamflow generation and cessation, and emphasize the need for spatially distributed modelling or monitoring of both
snowpack and rainfall dynamics.
**Keywords**: snowfall fraction, SWI, SWE, recharge, streamflow, dry-out date, non-perennial, satellite, iSnobal

**1. Introduction**
Due to increases in temperature, mountainous regions will receive less snowfall and more rainfall (Barnett et al., 2005; Stewart,
2009). This will alter the timing and amount of snowmelt, a significant source for water resources across the globe (Barnett et
al., 2005; Marks et al., 1999; Somers and McKenzie, 2020; Viviroli et al., 2007). On the scale of the continental United States
(US), a decrease in the fraction of precipitation falling as snow (snowfall fraction hereinafter) is expected to decrease stream

discharge (Berghuijs et al., 2014). Earlier stream discharge peaks in response to earlier snowmelt and a decline in summer low flows across the semi-arid mountainous US have been reported in both observational data records (McCabe et al., 2017; Luce and Holden, 2009; Regonda et al., 2005) and future climate projections (Naz et al., 2016; Leung et al., 2004; Milly and Dunne, 2020; Christensen et al., 2004). However, lower snowfall fractions in much of the western United States have not yet led to a significant decrease in annual discharge (McCabe et al., 2017). Understanding year-round discharge responses, and in particular the sensitivity of stream discharge to changes in yearly snowfall fractions, is therefore warranted, and will help us to anticipate how stream discharge might be affected by climate change.

Variations in snowfall fractions can affect the temporal distribution of surface water inputs (SWI = rainfall + snowmelt onto the soil). Snowpacks store water and release snowmelt later, whereas rain on snow-free ground immediately enters the hydrologic system. After rainfall or snowmelt reaches the ground surface, it might become stream discharge, remain stored on the land surface or in the soil, recharge deeper groundwater, or become evaporated or transpired. Generally, water inputs from rain or snowmelt during periods with high antecedent wetness and low evapotranspiration rates are more likely to recharge groundwater and generate discharge (Jasechko et al., 2014; Molotch et al., 2009; Hammond et al., 2019). Rainfall and snowmelt inputs might result in similar runoff ratios (discharge/SWI) as long as the overall catchment wetness is similar or if the catchment is wet at key locations for water transport (Seyfried et al., 2009). These antecedent wetness conditions may reflect the ability of catchment storage to provide a "memory effect" of past inputs that can buffer short-term changes in inputs. However, this memory varies among catchments and across years (depending, for instance, on the local subsurface storage capacity or yearly variations in evapotranspiration), so that changes in snowfall fractions might not always affect stream discharge. Prevailing climatic conditions and subsurface storage capacity might also influence how precipitation is partitioned takes after it reaches the ground surface (Hammond et al., 2019), indicating that both the temporal and spatial distribution of SWI are important when considering how snowfall fractions affect seasonal to annual stream discharge generation.

Snowfall fractions may also influence the spatial distribution of SWI. In the semi-arid western US, rainfall magnitudes generally increase with elevation (Johnson and Hanson, 1995). In regions with large snowfall fractions, this general elevation-driven pattern can be overlain by impacts of wind-driven redistribution of snow, which is dependent on factors such as topography, aspect, wind speed and wind direction (Sturm, 2015; Tennant et al., 2017; Winstral and Marks, 2014; Trujillo et al., 2007). Hence, differences in the SWI distribution due to varying snow depths could be particularly substantial in areas where wind-driven redistribution of snowfall is significant. The primary controls (e.g., topography, aspect, elevation) on snow depth and snow water equivalent (SWE) are relatively consistent from year to year, so the interannual distribution of snow is usually spatially consistent (Parr et al., 2020; Sturm, 2015; Winstral and Marks, 2002). The effects of elevation and aspect on the spatial distribution of snow depth, and thus the potential for SWI as snowmelt, are well-studied in both high and mid-altitude mountains (e.g., Grünewald et al., 2014; López-Moreno and Stähli, 2008; Tennant et al., 2017). Studies on snow drifting in seasonally snow-covered areas (Mott et al., 2018), prairie and arctic environments (e.g., Fang and Pomeroy, 2009; Parr et al., 2020) and in the context of avalanches (e.g., Schweizer et al., 2003), have shown that snow drifts can strongly influence the spatial water balance. These studies also revealed that equator-facing slopes might only receive half as much SWI as snowmelt compared to snow drift areas (Flerchinger and Cooley, 2000; Marshall et al., 2019). In turn, water originating from snow drifts can locally control groundwater level fluctuations (Flerchinger et al., 1992), and contribute to streamflow into the summer season (Chauvin et al., 2011; Hartman et al., 1999; Marks et al., 2002). The relative importance of spatial snowmelt patterns is expected to increase with snowmelt magnitude, which is sensitive to snowfall fractions. Hence, quantifying spatial snowmelt patterns in areas that are not seasonally snow-covered, and determining the importance of snow drifts for streamflow generation in these areas, could be an important step in clarifying how stream discharge is affected by snowfall fractions.

One area where the snowfall fraction varies substantially from year to year is the rain-snow transition zone. The rain-snow transition zone is an elevation band in which the dominant phase of winter precipitation shifts between snow and rain (Nayak et al., 2010), and is often characterized by a transient snowpack in (at least) parts of the defined area. Multiple studies in the European Alps and the north-western US have shown that snowfall fractions in the rain-snow transition zone are particularly vulnerable to increases in temperature associated with climate change (e.g., Stewart, 2009). For example, the snowfall fraction in the Swiss Alps is projected to decrease between 50% (at ~2000 m) to 90% (at ~1000 m) towards the end of the century (Beniston et al., 2003). The current extent of the rain-snow transition zone covers about 9200 km$^2$ in the Pacific Northwest of the US alone (here defined as Oregon, Washington, Idaho and the western part of Montana; Nolin and Daly, 2006), and is expanding and moving to higher elevations in response to climate change (Bavay et al., 2013; Nayak et al., 2010). This migration of the transition zone can affect precipitation patterns as well as discharge generation and timing across mountain ranges, with notable effects at the elevations surrounding the transition zone.

In addition to the expected decrease in snowfall fractions with climate change, annual climate variations are expected to increase almost everywhere across the planet (Seager et al., 2012), affecting annual runoff efficiency (Hedrick et al., 2020) and likely also influencing stream discharge timing and magnitude. One well-documented discharge response is that years in which catchments receive less snow have earlier snow-driven discharge peaks (McCabe and Clark, 2005; Stewart et al., 2005). This is relevant because earlier spring snowmelt has been linked to an increased risk of wildfire for catchments across the western US (Westerling et al., 2006), as well as to earlier and lower low-flows in late-summer and fall months (Kormos et al., 2016). In some catchments and years, portions of the stream network might also completely cease to flow and this drying can alter the network's ecological and biogeochemical functioning (Datry et al., 2014). In mid-elevation rain-snow transition zones, the annual snowpack variability is already relatively large. For example, in the Reynolds Creek Experimental Watershed (RCEW, in Idaho, US) the coefficient of variation (CV) of peak snow-water equivalent (SWE) between 1964 and 2006 ranged from 0.28-0.37 for five high-elevation stations (2056-2162 m) and was 0.72 for a mid-elevation weather station at the rain-snow transition zone (1743 m, Nayak et al., 2010). This mid-elevation variability suggests that year-to-year differences in snowfall at the rain-snow transition zone might already be substantial compared to nearby catchments at higher elevations, and allows for the investigation of catchment hydrologic responses to snowfall variations using a relatively short data record, especially in sites where precipitation inputs are likely to be strongly reflected in stream discharge (i.e., with a limited memory of past inputs). Using observations of hydro-climatically different years (e.g., rainy vs. snowy) could reveal how discharge and stream drying at the rain-snow transition zone has responded to past variations in water inputs, and thereby provide insight in how other small (<10 km$^2$) catchments with a similar vegetation cover and precipitation regime might respond to future changes in rain/snow apportionments.

Thus, the overarching goal of this work is to improve our understanding of discharge responses to year-to-year variations in precipitation phase and magnitude. We do this at the rain-snow transition zone - a region that experiences large year-to-year variations in snowfall fractions, covers a significant part of the land surface and might extend to higher elevations due to climate change. Specifically, we address the following research questions:

1.  How does the spatial and temporal distribution of SWI at the rain-snow transition zone vary between particularly wet, dry, rainy or snowy years?
2.  How does stream discharge timing and amount respond to SWI in wet, dry, rainy or snowy years?
3.  Are variations in stream discharge related to variations in yearly snowfall fractions?

Examining natural variation in snowfall fractions in the rain-snow transition zone contrasts with other research on snow-related processes that focus on seasonally-snow covered catchments. While many studies of snowmelt runoff examine seasonal

responses at the landscape scale, here we focus on hourly responses at a fine spatial resolution. This allows us to investigate the spatial distribution of the snowpack and snowmelt, as well as the phase of precipitation and the temporal distribution of SWI. Furthermore, while SWE is frequently used as a summarizing variable for winter precipitation when comparing precipitation to stream discharge, SWI is more directly related to the timing and amount of water resources, and might therefore be an important variable to model in future work addressing similar questions.

## 2. Site description

Our study location is Johnston Draw, a 1.8 km$^2$ headwater catchment at the Reynolds Creek Experimental Watershed (RCEW) in Idaho, USA. Elevations range from 1497 to 1869 m a.s.l., and mean annual air temperature and precipitation are 8.1 °C and 609 mm, respectively (2004-2014; Godsey et al., 2018). Previous research in RCEW has shown that mid-elevation catchments (1404 and 1743 m a.s.l.) have seen an increase in minimum daily temperatures (+0.57°C/decade), reduced snowfall (-32 mm/decade), and a decrease in streamflow (-13.8 mm/decade) over the 1965-2006 data record, while there was no change in total precipitation (Nayak et al., 2010; Seyfried et al., 2011). These streamflow trends are unlikely to be driven by increased plant water use (caused by increased temperatures) because there is only a short time window (~weeks) in which plant leaf-out has occurred and there is still sufficient soil water available in this water-limited environment (Seyfried et al., 2011). The catchment is underlain by granite bedrock (79%), with some basalt (3%) and tuffs (18%) (Stephenson, 1970), and slightly deeper soils exist on the north-facing slopes, although the difference is not significant (1.31±0.56 m vs. 0.77±0.34 m, respectively, p-value: 0.05; Patton et al., 2019). Annual average soil water storage on the north-facing slopes is larger than on the south-facing slopes, which is largely due to the difference in soil depth and a later start of vegetation growth compared to south-facing slopes (Godsey et al., 2018; Seyfried et al., 2021). Snowberry (*Symphoricarpos*), big and low sagebrush (*Artemisia tridentate* and *Artemisia arbuscula*), aspen (*Populus tremuloides*) groves and wheatgrass (*Elymus trachycaulus*) characterize the north-facing slopes, whereas the south-facing slopes host *Elymus trachycaulus*, *Artemisia arbuscula,* mountain mahogany (*Cercocarpus ledifolius*) and bitterbrush (*Purshia tridentate*) (Godsey et al., 2018). Discharge at the catchment outlet is non-perennial, and the stream at the catchment outlet typically flows from early November until mid-July (MacNeille et al., 2020).

## 3. Methods

### 3.1 Hydrometeorological and discharge data

We used hourly hydrometeorological data recorded at eleven weather stations throughout the catchment (Fig. 1; Godsey et al., 2018). The stations are placed at 50-m elevation intervals on the north and south-facing slopes, and span a ~300 m elevation range (1508-1804 m a.s.l.; see Marks et al., 2013 for a detailed description). Observations started in 2002, although some stations were placed only in 2005 or 2010, and some were decommissioned in 2017 (see Godsey et al., 2018 for exact years). Air temperature, solar radiation, vapor pressure and snow depth were measured at hourly intervals at each of the stations, and additional measurements of wind speed, wind direction and precipitation were available at jdt125, jdt124, and jdt124b. The snow depth time series were processed to remove gaps and unreliable measurements during storms and smoothed over an 8-h window in most cases, and a 40-h window under specific circumstances (Godsey et al., 2018). Stream discharge data (Godsey et al., 2018) were obtained with a stage recorder using a drop box weir at the watershed outlet (Pierson et al., 2000). Stage height was converted to discharge using a rating curve (Pierson and Cram, 1998), and discharge was frequently measured by hand to ensure high data quality (Pierson et al., 2000).

### 3.2 Remotely sensed observations

To characterize the spatial distribution of snow depth, a 1-m resolution snow depth product was calculated as the difference between a snow-off LiDAR flight (10-18 November 2007; Shrestha and Glenn, 2016) and a snow-on LiDAR flight (18 March 2009, around the time of peak accumulation), hereafter referred to as lidar snow depth. Typical vertical accuracies for lidar surveys are ~10 cm (Deems et al., 2013). We assumed that uncertainties in both lidar surveys were uncorrelated, resulting in an overall uncertainty of ~14 cm for lidar snow depth (summation in quadrature of uncertainties associated with each survey). All pixels that yielded a negative snow depth were excluded. The lidar snow depths were higher than the weather station snow depths, but this pattern was consistent across the catchment resulting in a strong linear relation between the two individual sets of snow depth measurements ($r^2$: 0.88, Supplemental Fig. S1).

Because only one lidar observation was available near peak snow accumulation, we also characterized snow presence throughout the season by mapping the snow-covered area (SCA) using satellite-derived surface reflectance at 3-m resolution, which is available starting in 2016 (4-band PlanetScope Scene; Planet Team, 2018). This high-resolution imagery was critical for our analysis because snow drifts in the rain-snow transition are relatively small in extent. No high-resolution satellite imagery was available for years that exhibited the key characteristics we sought to study (e.g., rainy, snowy, wet or dry; see section 3.3), so we focused on the most recent snow-covered period for which streamflow data and Planet imagery were available (1 November 2018 until 31 May 2019) to assess snow coverage. This targeted year was warmer than the year for which the lidar observations were available (mean annual air temperature: 8.0°C compared to 6.7°C in 2009), which may have resulted in earlier peak streamflow, melt-out date, and dry-out date for the stream. We manually selected all available images in which the entire watershed was captured and for which snow was visually recognizable, then removed all images for which clouds significantly covered the watershed, resulting in 41 usable images. The information from all four spectral bands was then condensed to one layer using a principal component analysis ('RSToolbox' package in R). We used the Maximum Likelihood Classification tool in ArcGIS (Esri Inc., 2020) to identify the SCA, after manually training the tool by selecting areas with and without snow cover (mean of 26895 pixels per class; median: 9019), visually aided by the original satellite imagery. Obtaining training data was most challenging during periods in which almost the entire area was snow-free or snow-covered, for densely vegetated areas, and when part of the catchment was shaded. To overcome the latter, we classified "snow-free", "snow-covered", and "shaded snow" in heavily shaded images, and afterwards merged "snow-covered" and "shaded snow". The mean confidence for all classifications is shown in Supplemental Fig. S2. Our method differs from other satellite-derived snow products that combine both visible and infrared light, but yielded a higher resolution data product (3-m resolution vs. 30-m for Landsat-8 or 500-m for MODIS) that was necessary to capture the snow drifts in the rain-snow transition zone.

We also used the surface reflectance imagery to determine the melt-out date of the snowpack for all years in which satellite and discharge observations were available (2016-2019). This was done by manually reviewing all available images and visually determining when all snow had melted. Given the high visiting frequency and limited cloudiness in early summer, we estimate that an error of ~2 days is appropriate for these melt-out dates.

**3.3 Spatially distributed snowpack modelling**

We used the Automated Water Supply Model (AWSM; Havens et al., 2020) to obtain spatially continuous estimates of the distribution and phase of precipitation, snowpack characteristics and surface water inputs (SWI). The two major components of AWSM are the Spatial Modeling for Resources Framework (SMRF; Havens et al., 2017) and iSnobal (Marks et al., 1999). SMRF was used to spatially distribute precipitation and all other weather variables (air temperature, solar radiation, vapor pressure, precipitation, wind speed and wind direction) along an elevation gradient using the hourly measurements from the weather stations. We included precipitation measurements from two stations within the basin (jdt125 and jdt124b) and two stations outside of the basin (jd144 and jd153, Fig. 1) to capture the elevation gradient. Precipitation at the wind-exposed site,

jdt124, was excluded because of precipitation undercatch issues. The interpolated vapor pressure and temperature fields were
then used within SMRF to calculate the dew point, and further distinguish which fraction of precipitation fell as rain and/or
snow. The output from SMRF was then used to force iSnobal, a physically-based, two-layer snowpack model that accounts
for precipitation advection from rain and snow (Marks et al., 1999).
The model was run at a 10-m resolution for five water years, namely, 2005, 2009, 2010, 2011 and 2014. We selected 2009
because the snow depth lidar survey was available in this year, and 2005, 2010, 2011 and 2014 because they are
hydroclimatically different. 2005 was rainy (snowfall fraction: 63% of the 2004-2014 mean and 23% of 2005 total
precipitation) whereas 2010 was snowy (snowfall fraction: 155% of 2004-2014 mean and 57% of 2010 total precipitation).
2014 was dry (precipitation: 86% of 2004-2014 mean) and 2011 was wet (precipitation: 132%) (Table 1, Supplemental Table
S3 and Fig. S4) with snowfall fractions of 41% and 30% of the total precipitation for each year, respectively. The work was
limited to four years because we aimed to focus on differences in the distribution of SWI and stream discharge for years that
had different snowfall fractions and total precipitation magnitudes. Therefore, these strongly contrasting years were selected
from 11 potential years of record (Godsey et al., 2018). Towards the end of the 2004-2014 period, more stations were deployed,
yielding additional observations to force the model with meteorological inputs and validate the model output of snow thickness,
so if conditions were similar, we selected later years within this period. We focus on the four hydroclimatically distinct years
in the results and discussion of this manuscript, but evaluate the model performance for all five years.
In order to represent the spatial variability in snowfall and the effects of wind redistribution of snow, we use the precipitation
rescaling approach proposed by Vögeli et al. (2016) that implicitly captures the spatial heterogeneity induced by these
processes using distributed snow depth information (e.g., from lidar or structure from motion (SfM)). This methodology can
be used to rescale the precipitation falling as snow to reproduce the observed snow distribution patterns while conserving the
initial precipitation mass estimation. Given the inter- and intra-annual consistency of spatial patterns of snow distribution
(Pflug and Lundquist, 2020; Schirmer et al., 2011; Sturm and Wagener, 2010), Trujillo et al. (2019) has been extending the
original implementation to utilize historical snow distribution information to other years in the iSnobal model. Following these
successful implementations, we used the spatial distribution of snow depth from the 2009 survey around peak snow
accumulation to inform the snowfall rescaling to all years in the study period. Although using the 2009 survey to rescale
snowfall in other years might have induced some uncertainty, verification of the interannual consistency in the snow
distribution in this catchment by comparing the lidar snow depth and the satellite imagery indicated that this uncertainty is
likely to be small.
**3.4 SWI**
One of the model outputs from iSnobal is 'surface water input' (SWI), which represents snowmelt from the bottom of the
snowpack, rain on snow-free ground, or rain percolating through the snowpack. Rainfall is directly counted as SWI when it
falls over snow-free ground, and it is included in the energy and water balances when it falls onto the snowpack. To calculate
snowmelt, iSnobal solves each component of the energy balance equation for each model time step using the best available
estimations of forcing inputs. Melt occurs in a pixel when the accumulated input energy is greater than the energy deficit (i.e.
cold content) of the snowpack. If the accumulated energy input is smaller than the energy deficit, the sum of current hour melt
and previous hour liquid water content will be carried over into the next hour. If that hour's input energy conditions are
negative, the liquid mass is refrozen into the column. Sublimation and evaporation of liquid water from the snow surface and
condensation of liquid water onto the snow surface is computed as a model output term, though these quantities were not
considered here. Canopy interception must be handled a priori when developing the model forcing input, and it was also not
considered here. Although not accounting for the latter introduces some uncertainty, we expect this to be small with the shrub
and grass vegetation types in Johnston Draw. Lastly, iSnobal is limited to snow processes only, which means that SWI 'exits'
the modelling domain. In reality, SWI either travels to the stream as surface or subsurface runoff, could be stored in the soil
until it evaporates or is transpired, or could recharge deeper groundwater storages. The route that SWI takes depends on the
overall catchment wetness as well as the local energy balance (e.g., incoming radiation) and vegetation activity. In this
manuscript, we computed SWI for each pixel and time step and assumed that all SWI generated in simulated snow-free pixels
was rain and that all SWI generated in simulated snow-covered pixels was snowmelt.

## 3.5 Model evaluation

Model results were evaluated in two ways. First, the simulated snow depths were compared to lidar snow depths covering the
entire basin on March 18, 2009; and second, the temporal variation of the simulated snow depths were compared to snow
depths measured at each of the weather stations for all simulated years. The latter comparison was done using model results
from a 30-m x 30-m area surrounding each station; this is equivalent to 3x3 grid cells because the model was run at a 10-m
resolution. We computed the Root Mean Square Error (RMSE) and Nash-Sutcliffe Efficiency (NSE; Nash and Sutcliffe, 1970)
for the observed versus simulated snow depths, as well as the NSE for the normalized observed versus normalized simulated
snow depths ($NSE_{norm}$). $NSE_{norm}$ reflects the ability of the model to reproduce the dynamic behaviour of the snowpack.

## 3.6 Comparison with discharge

The phase and magnitude of precipitation and the magnitude and temporal distribution of SWI were compared to annual
discharge and the stream dry-out date. The stream dry-out date is the day when the stream first ceased to flow at the catchment
outlet. For comparisons across seasons, we defined winter as December, January and February; spring as March, April and
May; summer as June, July and August, and fall as September, October and November. To compare SWI with the dry-out
date, we also calculated how much SWI occurred during the water year before the stream dried. No delays were considered
when comparing SWI to discharge (e.g., discharge as a fraction of SWI in January results from dividing discharge in January
by SWI in January). Discharge metrics were also compared to the flashiness of SWI inputs, which was calculated as the sum
of the difference in total SWI from day to day, divided by the sum of SWI (also known as the Richards-Baker Flashiness Index;
Baker et al., 2004). Further metrics included the fraction of time that more than half of the catchment was snow-covered and
the melt-rate between 40% snow-coverage in the catchment and the date at which the catchment was snow-free. A threshold
of 40% snow-coverage was chosen because this resulted in an approximately linear melt-rate for all years.

## 4.  Results

### 4.1 Snow depth observations

The lidar snow depth ranged from 0 to 5.3 m on the date of acquisition (18 March 2009), which was near peak snow cover
(median: 0.4 m; CV: 0.91; Fig. 2a). The south-facing slopes had little to no snow cover (mean: 0.3 m), whereas the north-
facing slopes were covered with 0.7 m of snow on average. For the years studied here, during the approximate duration of the
snowy season between 15 Nov and 15 Apr, the average snow depth for all north-facing stations was more than five times that
of the average snow depth at south-facing stations (0.20 vs. 0.04 m, respectively), and the snowpack lasted almost 90 days
longer on average (132 vs. 43 days, respectively). Weather stations on north-facing slopes and at higher elevations generally
had deeper snowpacks and were snow-covered longer than sites on the south-facing slopes or at lower elevations (Godsey et
al., 2018). The snowpack distribution was also affected by wind-driven redistribution of snow. For instance, snow depths at
jdt2 (north-facing) and jdt3b (south-facing) were consistently lower than at the weather stations directly below them in

elevation (jdt1 and jdt2b, respectively). Large snow drifts formed in some western parts of the watershed, up to a maximum depth of 5.3 m (90th percentile of all snow depths = 1.2 m, Fig. 2a). Wind-driven redistribution of the snow in Johnston Draw is facilitated by a relatively consistent southwestern wind direction (average during storms: 225°), and high wind speeds (average during storms at wind-exposed station jdt124: 6.7 m s$^{-1}$; Supplemental Fig. S5).

## 4.2 Snow depth model performance in space and over time

Simulated snow depths on the day of the lidar survey agreed well with the lidar snow depth ($r^2$: 0.88, Fig. 2a-c). The residual snow depths (lidar – simulation) were approximately normally distributed, with a mean of 0.2 m (see Supplemental Fig. S6 for a histogram and Q-Q plot). The largest differences (maximum difference: 1.1 m) between the simulated and measured snowpack were for isolated 10 m pixels on both the north- and south-facing slopes (Fig. 2a-c). The spatial pattern of the lidar snow depth also agreed well with the spatial patterns of snow-covered area (Fig. 2a,d), and there was a strong agreement between the simulated snow-covered area for 2009 (Fig. 2e) and the snow-covered area determined from satellite imagery for 2019 (Fig. 2d), including the modelled duration of snow cover and the number of satellite images in which snow-covered areas were observed. The largest discrepancy between the simulated and imagery-based snow duration was in the scour zone west of the snow drifts, where the model underestimated snow duration. Nonetheless, the consistent locations of the snow drifts between 2009 (observed and simulated) and 2019 (observed) indicates that the model captured the spatial distribution of the snowpack.

The median NSE for the hourly simulated snow depths compared to observations at the weather stations ranged from 0.22 (wet 2011) to 0.86 (snowy 2010) for all modelled years and weather stations, with RMSE ranging from 0.008 to 0.097 m (Table 2, see Supplemental Fig. S7 for time series of all simulated and observed snow depths). RMSE was equal to or lower than 0.1 m for all years, with the year in which the NSE performance was lowest (wet 2011) having an RMSE of 0.046 m. The temporal variation of the snowpack at each of the weather stations was well-captured by the model; the median NSE for the normalized snowpack depths (NSE$_{norm}$) ranged from 0.65 to 0.94 (median: 0.75), although there were some sites and years with low NSE (Table 2). Both high and low NSE values are observed at nearly all of the stations (e.g., range NSE at jdt4: -9.60 to 0.91 and jdt1: 0.01 to 0.83) with lower values at some sites in 2011. Possible explanations for the relatively low performance at the remaining sites are discussed further in section 5.3. Despite the low performances for some years and locations, the normalized snow depths were largely acceptable (35 out of 40 year/location-combinations had NSE$_{norm}$ value above 0.5; Table 2). The generally strong performance lends confidence that the simulation of ablation and accumulation processes in the model is reasonable and implies that the temporal distribution of snow-covered area (SCA) and surface water inputs (SWI) simulated by the model are reliable.

## 4.3 Spatial and temporal pattern of surface water inputs (SWI)

The spatial pattern of SWI was similar for all years, with the highest SWI occurring in the snow drifts (maximum SWI (SWI$_{max}$): 3892 mm; 98th percentile of SWI (SWI$_{98}$): 1235 mm, both in wet 2011; Fig. 3, Table 1). Annual SWI across the rest of the catchment varied less, with north-facing slopes receiving 45 to 127 mm more SWI than south-facing slopes (values for rainy 2005 and snowy 2010, respectively; Table 1). Snow drift locations received 1.7 to 2.7 times more SWI than the catchment average (ratio SWI$_{98}$/SWI$_{avg}$). Summarizing SWI by aspect (see polar diagrams in Fig. 3) revealed the highest SWI on northeast-facing slopes and roughly equal annual SWI for all other aspects. Differences between the northeast-facing slopes and other parts of the catchment were largest in snowy 2010 (ratio of major/minor axis of polar plot: 1.29), and smallest in rainy 2005 and dry 2014 (ratio: 1.13 and 1.17, respectively).

Weekly sums of SWI ranged from 0 to ~75 mm in all years (Fig. 4). Summer most frequently had weeks without SWI generation, whereas the highest weekly SWI occurred with simultaneous rainfall and snowmelt (i.e., rain-on-snow events, such as the one visible in February 2014, Fig. 4d). However, large rainfall events without snowfall or snow cover in spring of rainy 2005 (weekly SWI: ~75 mm) and in fall of wet 2011 (weekly SWI: ~50 mm; grey peaks in Fig. 4a and c) also generated high SWI. In 2011, the majority of SWI was generated in winter and spring (47% between December and May, see inset in Fig. 4c) whereas in dry 2014 most SWI was generated in winter (54% between December and February, Fig. 4d). In 2005 and 2010, most SWI was generated in spring (March-May 32% and 46%, respectively). Similar amounts of SWI occurred in spring in 2005 and 2010 (339 and 388 mm, respectively); however, in 2005, 93% came from rain whereas in 2010, only 35% came from rain. Average daily spring SWI rates were higher in snowy 2010 than in rainy 2005 (mean spring SWI rate: 3.7 mm $d^{-1}$ in 2010 vs. 2.9 mm $d^{-1}$ in 2005). Overall, variations in weekly and daily SWI rates were lower in snowy 2010 (CV daily SWI: 1.71) than in all other years (2.50 in 2005, 2.14 in 2011, and 2.65 in 2014).

### 4.4 Stream discharge

Streamflow was least responsive to SWI at the beginning of each water year (Fig. 5). For instance, in 2005 and 2010, 174 and 108 mm of SWI occurred before February $1^{st}$ (31% and 20% of annual SWI), whereas discharge amounted to only 7% and 1% of its yearly total during that same period. Similarly, 82 mm of SWI in October 2011 resulted in less than 1 mm discharge, whereas 180 mm of SWI in Nov-Jan led to 62 mm of discharge. SWI generally resulted in most discharge when SWI rates were high, such as during a 3-day rain-on-snow event in February 2014 (SWI: 75 mm, discharge: 29 mm) or during spring snowmelt in April 2011 (SWI: 108 mm, discharge: 102 mm). Such individual precipitation events had a strong influence on the annual runoff efficiency. For instance, 2014 had a slightly higher runoff efficiency (0.16) than 2005 (0.11) and 2009 (0.14), mostly due to the high runoff generation during one rain-on-snow event (29 mm, 36% of yearly discharge).

Annual discharge was highest in 2011 (307 mm, 43% of SWI) and lowest in 2005 (62 mm, 11% of SWI). Despite similar SWI in 2005 and 2010 ($SWI_{avg}$: 553 and 557 mm, respectively, Table 1), snowy 2010 had nearly twice as much annual discharge as rainy 2005 (117 mm or 21% of SWI vs. 62 mm or 11% of SWI, respectively). Apart from these two years, there was no relation between annual discharge and the annual snowfall fraction (Fig. 6c), nor between annual discharge and the amount of SWI produced by rainfall or snowmelt in different seasons (winter, spring, summer, or any combination of these periods). By considering additional years (for which SWI was not simulated), we found that annual discharge was positively related to the amount of precipitation recorded at the lowest elevation precipitation station (jdt125, $r^2=0.83$, Fig. 6a). Annual discharge was slightly higher for years that were preceded by a year that received above average annual precipitation (see Supplemental Fig. S8), but the correlation coefficient decreased when including the precipitation totals recorded in the preceding year (e.g., annual discharge vs. precipitation in the same year + 0.5 times precipitation previous year). This indicates that any memory effect is likely to be small in this catchment. Frequent stream drying (16 out of 18 years between 2003 and 2020, data not shown, the stream did not cease flow in 2006 and 2011) and the high potential evaporation rates in this semi-arid, high desert system (evapotranspiration accounts for nearly 90% of precipitation in the nearby Upper Sheep Creek catchment; Flerchinger and Cooley, 2000) also suggest that any water in the shallow, 'active' subsurface storage is likely limited, and that any memory effect, if present, is perhaps constrained to deeper subsurface water storages.

Comparison of annual discharge and the stream dry-out date to metrics describing the phase and magnitude of precipitation, the temporal distribution of SWI and key characteristics of the snowpack highlighted the importance of the magnitude and timing of SWI (Fig. 7). Significant relationships with annual discharge were found for annual precipitation (Fig. 6a) and the sums of precipitation and snowfall in spring (Fig. 7 and Supplemental Fig. S9). The dry-out date of the stream was significantly correlated to annual precipitation, the sum of winter and spring precipitation and spring snowfall, spring precipitation as a

fraction of SWI, the melt-out date of the snowpack, and the sum of SWI before the dry-out date (Fig. 7 and Supplemental Fig.
S9). No significant correlation was found between the annual, winter and spring snowfall fraction and annual discharge and
the stream dry-out date (Fig. 7).

**5. Discussion**
**5.1 Spatial variability in SWI**
Snow drifting and aspect-driven differences in snow dynamics caused a strong variability in the spatial pattern of the snowpack
(Fig. 2a) and SWI (Fig. 3). We found that the spatial pattern in simulated SWI was similar across all years, with snow drifts
receiving up to seven times more SWI than the catchment average ($SWI_{max}/SWI_{avg}$ in 2010, Table 1). Even in rainy 2005, SWI
was more than 3.5 times higher in the snow drifts ($SWI_{max}$: 2005 mm) compared to the catchment average ($SWI_{avg}$: 573 mm,
Table 1). In our modelling routine, the spatial consistency between years is pre-determined by the snowfall rescaling (see
section 3.3), but this likely also reflects real-world conditions, as suggested by the spatial agreement between the independently
collected satellite imagery and lidar snow depths (Fig. 2). Most importantly, the nearly four-fold variation in SWI over less
than a kilometre distance is equivalent to the average precipitation difference between most of Reynolds Creek and the peaks
of the Cascade Mountains in Oregon hundreds of kilometres away, or equivalently, shifting from a semi-arid steppe to coastal
mountain snowpack. This difference directly affects water-limited processes such as weathering or plant species distribution.
In Johnston Draw, this is clearly visible: aspen stands are located directly below snow drifts (Kretchun et al., 2020) and
sagebrush dominates the rest of the catchment. Because snow drifts drive the spatial pattern of SWI, it is crucial to quantify
wind-driven redistribution processes as well as capture aspect and elevation-driven processes, even at the rain-snow transition
zone.

Snow drifts delivered 4.2% (2005) to 7.2% (2010) of the basin-total annual SWI on just ~2% of the land surface, and snow in
drifts persisted longer, compared to non-drift areas, into the spring season (Fig. 2d-e). Previous work in the seasonally snow-
covered Reynolds Mountain East catchment showed that snow drifts indeed hold a large fraction of total catchment snow water
equivalent (SWE), with 50% of total SWE on just 31% of the catchment area (Marks et al., 2002), and SWI varying strongly
in space, ranging from 150 to 1100 mm for individual grid cells (10 – 20 m) in the relatively dry water year 2003 (Seyfried et
al., 2009). Snow drifts in Johnston Draw were shallower (up to 5 m in 2009) and covered a smaller portion of the area (~2%)
than in the higher elevation Reynolds Mountain East catchment, but are proportionally even more important in the rain-snow
transition zone by holding up to 15% of SWE during peak SWE in snowy 2010 and 25% in rainy 2005. Water originating from
snow drifts has been shown to locally control groundwater level fluctuations (Flerchinger et al., 1992), and contribute to
streamflow into the summer season (Chauvin et al., 2011; Hartman et al., 1999; Marks et al., 2002). For instance, in the Upper
Sheep Creek watershed, also in RCEW, Chauvin et al. (2011) showed that the lowest stream discharge was recorded for the
year in which snow drifts were least prominent. In Johnston Draw, the stream dry-out date was positively correlated with the
drift melt-out date (Fig. 6b), suggesting that isolated snow patches are also here important for sustaining streamflow. These
results do not reveal the mechanism or influence of the specific drift location since neither subsurface flow nor streamflow
generation processes were measured or simulated. Nonetheless, observations of snow drifts from high-resolution satellite
imagery are largely consistent with model simulations of SCA (Fig. 2) and thus may be used to predict stream drying in drift-
influenced watersheds.

## 5.2 Temporal variability in SWI and discharge response

We found that the majority of SWI occurred in winter and spring, and that catchment-average SWI was more uniform in time in snowy 2010 than in the other years (CV of daily SWI in 2010: 1.7; other years: 2.14 – 2.65). The steadier water inputs in the snowmelt period might explain why annual discharge in snowy 2010 was double that of rainy 2005 despite similar total SWI. More stable water inputs from snowmelt rather than flashy water inputs from rain could have led to wetter soils and higher soil conductivity rates, allowing more water to pass through the subsurface towards the stream or towards deeper storage (Hammond et al., 2019). Previous work in the nearby Dry Creek Experimental Watershed (Idaho) showed that water stored in the soil dries out approximately ten days after snowmelt (McNamara et al., 2005). For the years on record here, streamflow was sustained for a minimum of 59 days after the melt-out date (Table 1), even though SWI is generally low after June each year (Fig. 4). This underscores that it is likely that deeper flow paths contribute to the stream in early summer. This is also consistent with stream discharge being nearly unresponsive to SWI during the dry catchment conditions in the beginning of each water year (Fig. 5). During fall, subsurface water storage across the catchment is low, and any SWI during this period thus likely results in recharge or evaporation rather than stream discharge (Seyfried et al., 2021). Air temperature also has a small effect on the runoff efficiency, particularly in the summer season. The runoff efficiency, calculated as summer discharge divided by summer precipitation for the 2004-2014 record, was significantly correlated to summer air temperatures ($r^2$=-0.54, p value=0.08, Supplemental Fig. S10) whereas this relationship was insignificant on the annual scale ($r^2$= -0.43, p-value=0.217; Supplemental Fig. S11). This suggests that evapotranspiration, which is directly affected by the ambient air temperature, has some influence on runoff efficiency, despite the catchment being an overall water-limited environment. In winter, higher temperatures result in higher runoff efficiencies ($r^2$=0.48, p-value=0.131, Supplemental Fig. S10), which is likely due to faster melt-out and more saturated soils, as described above. However, further simulations are required to fully understand how precipitation amounts, timing and location interact with subsurface water storage to control stream discharge.

In contrast to our hypothesis and what has been suggested in the literature (e.g., based on the comparison of 420 catchments in the continental US using the Budyko framework, Berghuijs et al., 2014), neither annual discharge nor the stream dry out-date were correlated with snowfall fraction (Fig. 6, 7). Instead, annual discharge and the stream dry-out date were more correlated with total precipitation and the snowpack melt-out date. This highlights the importance of the temporal distribution of SWI, which is not captured in an annual snowfall fraction. The temporal distribution of SWI might be less important for predicting stream discharge and cessation in more humid catchments in which precipitation is more evenly distributed over the year and/or in which more precipitation events occur, or in larger catchments, such as those considered in Berghuijs et al., (2014; range catchment areas: 67-10,329 km$^2$). We found that individual precipitation events can also heavily influence the yearly runoff efficiency, as described for 2014 (section 4.4). As such, considering inter-annual variability and rainfall or snowmelt events is an important addition to annual average values when investigating how precipitation affects discharge in semi-arid regions.

Bilish et al. (2020) similarly found that streamflow was not correlated to the snowfall fraction for a small catchment with an ephemeral snowpack in the Australian Alps. They attributed this to the frequent occurrence of mid-winter snowmelt: the snowpack melted out several times each year, independent of the annual snowfall fraction, and the snowpack thus did not store a significant amount of water. Field observations at Dry Creek, a nearby semi-arid catchment that includes a rain-dominated and a snow-dominated area, also suggested that the snowfall fraction was not related to annual discharge for a small sub-catchment at the rain-snow transition zone (Treeline sub-catchment, 0.015 km$^2$), but snowfall fraction was correlated with annual discharge when considering the entire Dry Creek catchment (28 km$^2$, J. McNamara, personal communication). Another study at Dry Creek suggested that the snowfall fraction is less important than spring precipitation to satisfy evaporative demands of upland ecosystems (McNamara et al., 2005), emphasizing the importance of the temporal distribution of SWI for

other semi-arid catchments. For the years studied here, we found that streamflow was sensitive to spring precipitation and total
precipitation, but that the snowfall fraction did not significantly affect stream discharge (Fig. 6, 7).
**5.3 Limitations and opportunities**
Though the model adequately reproduced the spatial snowpack patterns and dynamics (Fig. 3 and Table 2), temporal variations
in the snow depths (i.e., melt and accumulation) recorded at the weather station locations were simulated better than the
absolute snow depths. To investigate why simulations of snow depths were poor for some stations and years, we calculated
the average and precipitation-weighted average wind directions, wind speeds and snow densities for all events during which
the snowfall fraction was higher than 0.2 (i.e., 20%; see Supplemental Table S12 and Fig. S13) from the station data. Although
wind speed and directions were generally consistent (Supplemental Fig. S13), in 2011, the combination of higher snow
densities (stronger cohesion of snow particles; 122 kg m$^{-2}$) and lower wind speeds (less energy for transport; 5.7 m s$^{-1}$)
compared to 2009 (102 kg m$^{-2}$ and 6.5 m s$^{-1}$, respectively, precipitation-weighted averages in Table S12) might have led to
less wind redistribution of snow in that year and correspondingly resulted in underpredictions of snow depths at north-facing
and high-elevation sites in 2011 (jdt3, jdt4, jdt5 and jdt124b). Since NSE values are based on squared errors, the divergence
between the simulated and observed snow depths impacted the model performance more severely in 2011 than in years with
shallower snowpacks (i.e., 2005 and 2014). The snowpack density, wind speed and wind direction values in 2005 diverged
most from 2009, from which the lidar observations were used, but nonetheless had a relatively high performance (NSE: 0.83),
possibly because there was data from only one station available for validation.
In addition to the uncertainty in the spatial redistribution of snow depending on wind speeds, wind direction and snow densities,
we suggest three additional reasons for the differences between simulated and observed snow depths. First, the varying
performance at jdt125 might be related to inaccuracies in calculating the phase of precipitation, which would most strongly
affect lower elevations at which the phase shifts more often from rain to snow. Any uncertainty in the magnitude or phase of
precipitation would decrease model efficiency because precipitation was interpolated based on elevation, after which the
proportion of precipitation falling as snow was redistributed based on the lidar snow depths (see section 3.3). Second, the
simulated snow depths reflect all processes occurring in each 10-m grid cell (our model resolution), whereas the ultrasonic
snow depth measurements represent processes at ~1-3 m$^2$. Small differences between the simulated and observed snow depths
are therefore expected. Third, iSnobal is a mass and energy balance model, and therefore optimized to correctly model mass.
Model evaluation using snow depths (instead of SWE) is thus less favourable, since small differences in snow densities and
SWE could lead to significant differences in snow depths. However, since snow depth measurements were available and SWE
measurements were not, we focused on snow depth. Uncertainties were also present in the weather station snow depths, as
well as the lidar-based snow depths and the satellite-based SCA analysis. We compared the spatial patterns from the lidar and
satellite imagery to test if the spatial pattern was consistent between these two data sources and found this to largely be the
case (Fig. 2). As such, we are confident that despite the uncertainties of our analysis, we captured the within-catchment
variability of the snowpack and also adequately modelled the variability in SWI that we set out to investigate.
Discrepancies between simulated and observed snow depths are challenging to solve, especially for areas with an ephemeral
snow cover (Kormos et al., 2014) or with complex vegetation patterns, such as the sagebrush in Johnston Draw. Shallow snow
covers are more sensitive to small variations in energy fluxes than deeper seasonal snow covers (Pomeroy et al., 2003; Williams
et al., 2009). As a result, small errors in the spatial extrapolation of the forcing data or in the forcing data itself (e.g., uncertainty
in the observed relative humidity or temperature) can introduce uncertainties in the model results (Kormos et al., 2014). For
instance, the transition from snow-covered to snow-free areas results in a large change in albedo, which influences solar
radiative fluxes. The snowpack at the rain-snow transition zone can melt out several times per year, even within a single day,

and melt-out dates are variable across the catchment. Therefore, a small error in the simulated melt-out date for each cell can result in a larger error in the basin-average or yearly results. Perhaps these challenges are also a reason for the limited number of studies that have simulated warm snowpacks (Kormos et al., 2014; Kelleners et al., 2010), despite multiple regional studies highlighting that the rain-snow transition zone is expanding and that their climates are changing rapidly (Klos et al., 2014; Nolin and Daly, 2006). Challenges linked to snow ephemerality likely also affected our results, but the agreement between the observed and simulated snow depths indicates that at least the general patterns of accumulation and melt in space and over time were represented by the simulations, at a scale that was small enough to characterize the snow drifts.

Regardless of the challenges that come with studying an intermittent snow cover, the relationship between the snowpack melt-out date and stream dry-out date poses interesting opportunities to inform hydrological models or evaluate model results with independent observations. Measurements of SCA can be obtained through satellite imagery and are thus easier and cheaper to obtain than SWE or snow depth measurements (e.g., Elder et al., 1991). Satellite observations can be particularly helpful to investigate remote areas that exceed a feasible modelling domain, and can be used to inform or evaluate models. Given the restrictions for satellite imagery imposed by clouds and visit-frequency, particularly for areas with an ephemeral snow cover that might melt out in a single day, a combination of satellite imagery and snowpack modelling seems a promising way to leverage these observations while ensuring the fine temporal resolution that might be needed to study stream cessation.

## 6. Conclusions

As a result of climate change, the rain-snow transition zone will receive more rain and less snow, which may influence the spatial and temporal distribution of surface water inputs (SWI, summation of rainfall and snowmelt). The goal of this work was to quantify the spatial and temporal distribution of SWI at the rain-snow transition zone, and to assess the sensitivity of annual stream discharge and stream cessation to the temporal distribution of SWI as well as to the annual snowfall fraction. To this end, we used a spatially distributed snowpack model to simulate SWI during five years, of which four had contrasting climatological conditions. We found that the spatial pattern of SWI was similar between years, and that snow drifting and aspect-controlled processes caused large differences in SWI across the watershed. Snow drifts received up to six times more SWI than other sites, and the difference between SWI from the snow drifts and catchment average SWI was highest for the year with the highest snowfall fraction. This highlights that the snowfall fraction affects the spatial variability in SWI, with more rain leading to less variability. The majority of SWI occurred in winter or spring, which was also the time that the percentage of SWI becoming streamflow was highest (up to 94% in April 2011). Over the 2004-2014 data record, annual discharge was insensitive to snowfall fraction and depended more on total and spring precipitation. The stream dry-out date was also sensitive to total and spring precipitation. In addition, stream cessation was positively correlated to the last day at which there was snow present anywhere in the catchment, which indicates that the persistence of snow drifts in small parts of the catchment is critical for sustaining streamflow. This study highlights the heterogeneity of SWI at the rain-snow transition zone and its impact on stream discharge, and thus the need for spatially and temporally representing SWI in headwater-scale studies that simulate streamflow.

**Data availability**

The hydrometeorological and discharge data used in this paper is available via Godsey et al. (2018), satellite imagery can be obtained via Planet Team (2018) and remaining data is available upon reasonable request.

**Author contribution**

LK developed the concept of the study together with SEG. LK, SH, ET, AH and KH performed and/or contributed to the simulations. LK prepared the first draft of the manuscript. All co-authors provided recommendations for the data analysis, participated in discussions about the results, and edited the manuscript.

**Competing interests**

The authors declare that they have no conflict of interest.

**Financial support**

This research has been supported by the Swiss National Science Foundation (grant no. P2ZHP2_191376) and the US National Science Foundation (award EAR-1653998). The Reynolds Creek Critical Zone Observatory Cooperative Agreement (EAR-1331872) provided support for processing the snow depth data.

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

**Tables**
**Table 1. Precipitation, discharge and SWI characteristics for each water year including: total precipitation (mm,**
**average of precipitation measured at jd124b and jd125), the fraction of precipitation falling as snow (snowfall fraction),**
**dates of the start (snow$_{start}$) and end (snow$_{end}$) of the snowy season, defined as > 1 cm of snow at weather station jdt124b**
**(except for 2005, for which only data for weather station jdt125 was available), dates at which the simulated snow cover**
**had melted (melt-out date; SCA = 0), annual discharge (Q$_{annual}$) and runoff efficiency (Q$_{annual}$/SWI$_{avg}$) as well as the**
**start (Flow$_{start}$) and end (Flow$_{end}$) of surface flow at the catchment outlet, and simulated surface water inputs (SWI).**
**We report the catchment-average SWI (SWI$_{avg}$) as well as SWI from rain (SWI$_{rain}$), SWI from snowmelt (SWI$_{snow}$), the**
**98$^{th}$ percentile of SWI (SWI$_{98}$), maximum SWI (SWI$_{max}$) and the average SWI for north-facing slopes (excluding the**
**drift area, SWI$_{NF-drift}$) and south-facing slopes (SWI$_{SF}$).**

| WY | | 2005 | 2009 | 2010 | 2011 | 2014 |
|---|---|---|---|---|---|---|
| | | Rainy | Lidar available | Snowy | Wet | Dry |
| Precipitation | mm | 542 | 549 | 531 | 693 | 450 |
| Snowfall fraction | - | 0.23 | 0.49 | 0.57 | 0.41 | 0.30 |
| Snow$_{start}$ | dd-mon (DOWY) | 16-Oct* (16) | 01-Nov (32) | 04-Oct (4) | 06-Nov (37) | 20-Oct (20) |
| Snow$_{end}$ | | 01-Mar* (152) | 19-Apr (201) | 26-May (238) | 01-May (213) | 06-Apr (188) |
| SCA = 0 | | 02-Jun (245) | 14-Jun (257) | 16-Jun (259) | 18-Jun (261) | 14-May (226) |
| Q$_{annual}$ | mm | 62 | 81 | 117 | 307 | 80 |
| Q/SWI$_{avg}$ | - | 0.11 | 0.14 | 0.21 | 0.46 | 0.16 |
| Flow$_{start}$ | dd-mon | 11-Nov (38) | 22-Nov (54) | 12-Nov (43) | 24-Oct (24) | 28-Oct (28) |
| Flow$_{end}$ | (DOWY) | 25-Aug (328) | 25-Aug (328) | 26-Aug (329) | - | 13-Jul (285) |
| SWI$_{avg}$ | mm | 557 | 587 | 553 | 672 | 506 |
| SWI$_{snow}$ | mm | 145 | 271 | 310 | 229 | 170 |
| SWI$_{rain}$ | mm | 412 | 316 | 243 | 443 | 336 |
| SWI$_{98}$ | mm | 982 | 1394 | 1513 | 1588 | 1015 |
| SWI$_{max}$ | mm | 2005 | 3350 | 3863 | 3892 | 2219 |
| SWI$_{NF-drift}$ | mm | 551 | 568 | 534 | 665 | 490 |
| SWI$_{SF}$ | mm | 505 | 456 | 407 | 556 | 430 |

**\*dates based on measurements at jdt125 (outlet) rather than 124b (close to top of the catchment, see Fig. 1)**

**Table 2. Nash-Sutcliffe Efficiency (NSE; Nash and Sutcliffe, 1970) and root mean square error (RMSE, m) for**
**simulated and observed snow depths at each weather station, as well as the NSE for normalized (z-transformed) snow**
**depths (NSE$_{norm}$). Dashes (-) indicate that no observed snow depths were available in that year. See Supplemental Fig.**
**S7 for the time series of observed and simulated snow depths.**

| | | Outlet | | North-facing | | | South-facing | | | Upper region | | Median |
|---|---|---|---|---|---|---|---|---|---|---|---|---|
| | **Station** | jd125 | jdt1 | jdt2 | jdt3 | jdt4 | jdt2b | jdt3b | jdt4b | jdt5 | jd124b | |
| **NSE** | **2005** | 0.83 | - | - | - | - | - | - | - | - | - | 0.83 |
| | **2009** | 0.45 | 0.67 | 0.09 | 0.95 | 0.91 | - | - | - | 0.65 | 0.84 | 0.67 |
| | **2010** | 0.01 | 0.92 | 0.91 | 0.68 | 0.86 | - | - | - | 0.67 | 0.92 | 0.86 |
| | **2011** | 0.40 | -0.46 | 0.63 | 0.03 | -9.60 | 0.52 | 0.76 | 0.54 | -0.06 | -5.56 | 0.22 |
| | **2014** | 0.80 | -2.07 | 0.76 | 0.49 | 0.25 | 0.39 | 0.60 | 0.80 | 0.81 | 0.66 | 0.63 |
| **NSE$_{norm}$** | **2005** | 0.87 | - | - | - | - | - | - | - | - | - | 0.87 |
| | **2009** | 0.65 | 0.50 | 0.50 | 0.83 | 0.85 | - | - | - | 0.89 | 0.97 | 0.83 |
| | **2010** | 0.25 | 0.94 | 0.92 | 0.96 | 0.95 | - | - | - | 0.68 | 0.94 | 0.94 |
| | **2011** | 0.86 | 0.34 | 0.73 | 0.89 | -0.86 | 0.55 | 0.75 | 0.67 | 0.63 | 0.15 | 0.65 |
| | **2014** | 0.77 | 0.59 | 0.75 | 0.81 | 0.64 | 0.33 | 0.64 | 0.72 | 0.80 | 0.79 | 0.74 |
| **RMSE (m)** | **2005** | 0.01 | - | - | - | - | - | - | - | - | - | 0.01 |
| | **2009** | 0.11 | 0.10 | 0.19 | 0.05 | 0.08 | - | - | - | 0.11 | 0.09 | 0.10 |
| | **2010** | 0.12 | 0.03 | 0.05 | 0.11 | 0.09 | - | - | - | 0.09 | 0.06 | 0.08 |
| | **2011** | 0.03 | 0.06 | 0.04 | 0.08 | 0.30 | 0.02 | 0.02 | 0.02 | 0.05 | 0.15 | 0.08 |
| | **2014** | 0.01 | 0.06 | 0.02 | 0.04 | 0.05 | 0.02 | 0.02 | 0.01 | 0.02 | 0.02 | 0.03 |


**Figures**

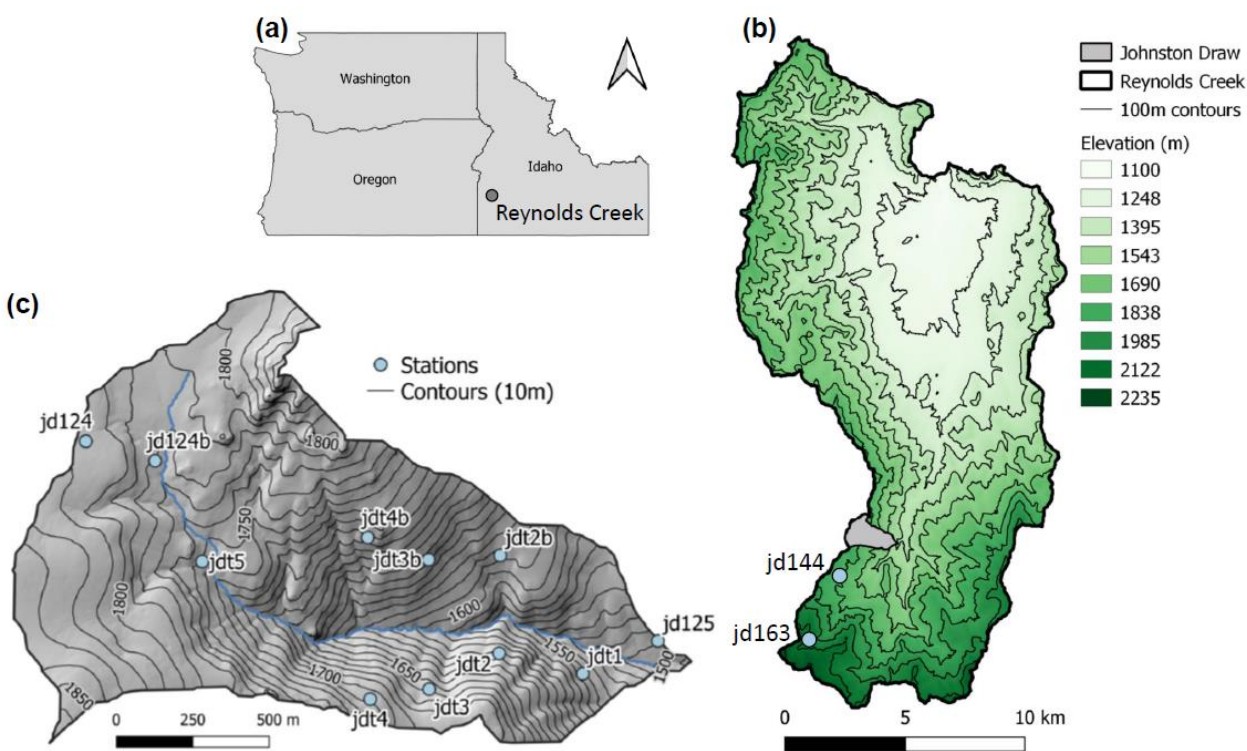

**Fig. 1 Maps of the location of (a) the Reynolds Creek Experimental Watershed (RCEW) in the state of Idaho (USA,**
**EPSG:4269 - NAD83 projection), (b) Reynolds Creek Experimental Watershed with indication of elevation (white =**
**lower, dark green = higher), 100 m contour lines, the location of Johnston Draw (grey polygon) and two additional**
**precipitation gauges (dots) indicated in light blue, and (c) Johnston Draw with the weather stations (light blue dots),**
**stream (blue line), and 10 m contour lines (black lines), overlain on a hillshade DEM.**

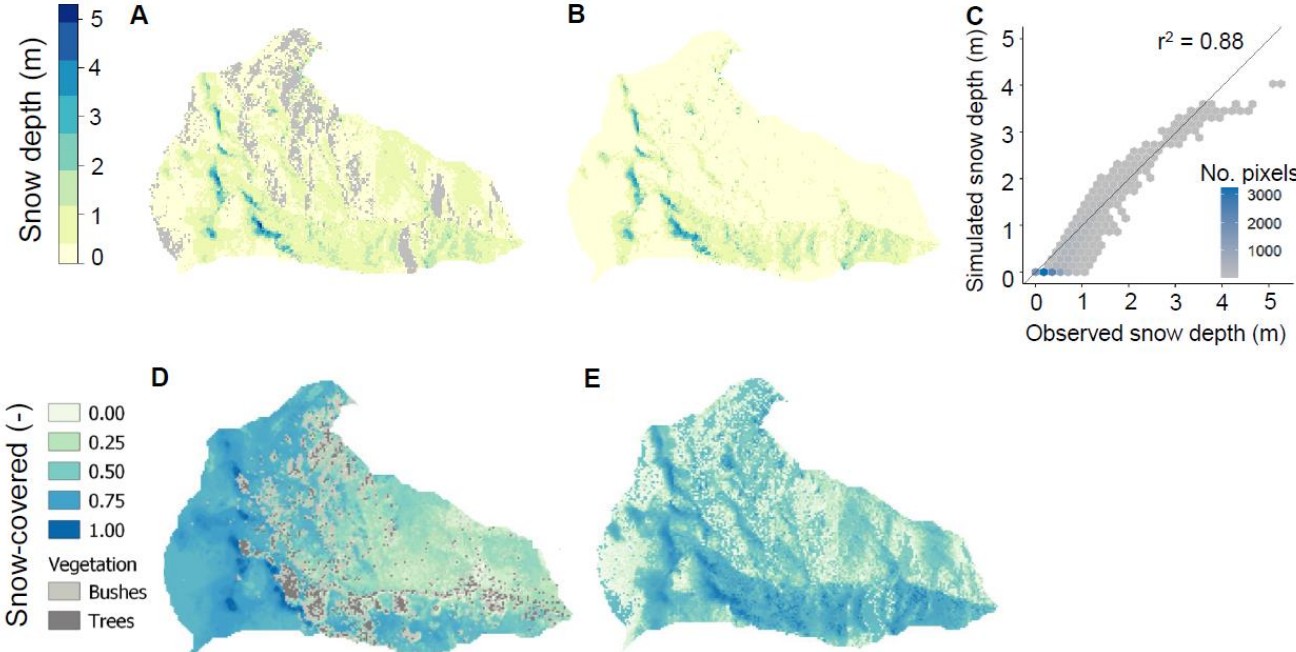

Fig. 2. (a) Lidar snow depth (m) at 3-m resolution on 18 March 2009, and (b) simulated snow depths for the same day, where yellow indicates low snow depths, blue high snow depths, and grey the areas for which the snow depth could not reliably be determined from the lidar measurement (see section 3.2). (c) Hexagonal bin plot comparing the observed and simulated snow depths with grey colors indicating fewer pixels and blue indicating more pixels included per bin. (d) Fraction of images for which sites were snow-covered, using 3-m resolution satellite imagery for the available images (n=41) of water year 2019 (see section 3.2), and (e) fraction of time during which each pixel was snow-covered, using the simulated snow cover from the beginning of the water year 2009 until all snow had melted (n=238). Bushes and trees (marked in grey in panel d) inhibited the exact determination of the snow cover for the satellite imagery in some locations.

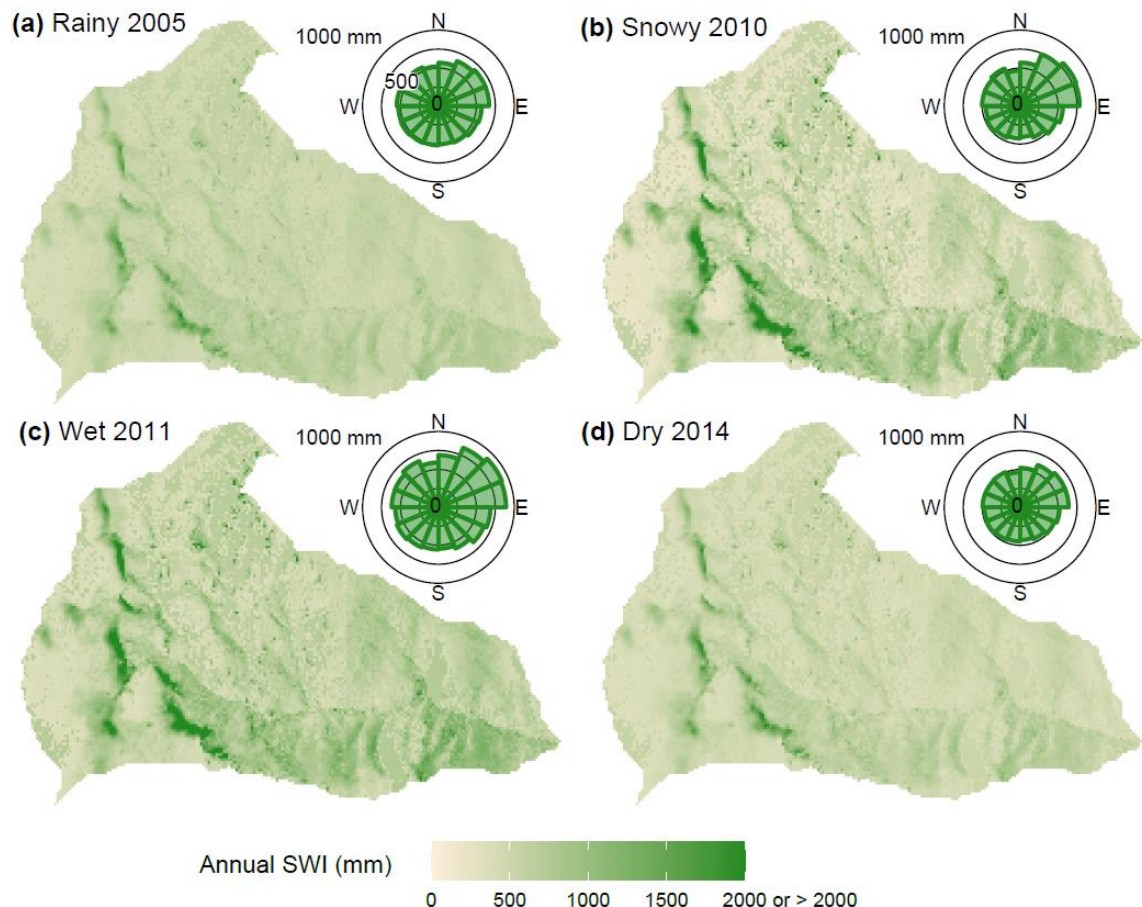

**Fig. 3. Maps showing the yearly sum of surface water inputs (SWI, mm) for (a) rainy 2005, (b) snowy 2010, (c) wet 2011**
**and (d) dry 2014, with polar diagram insets showing the average sum of SWI per 10-m grid cell for each aspect (binned**
**per 22.5°). Higher SWI values are shown in darker colours, lower SWI values in lighter colours, and SWI values are**
**capped at 2000 mm to enhance the contrast. Maximum annual SWI values are shown in Table 1 and a map of simulated**
**SWI for 2009 is shown in Supplemental Fig. S13.**

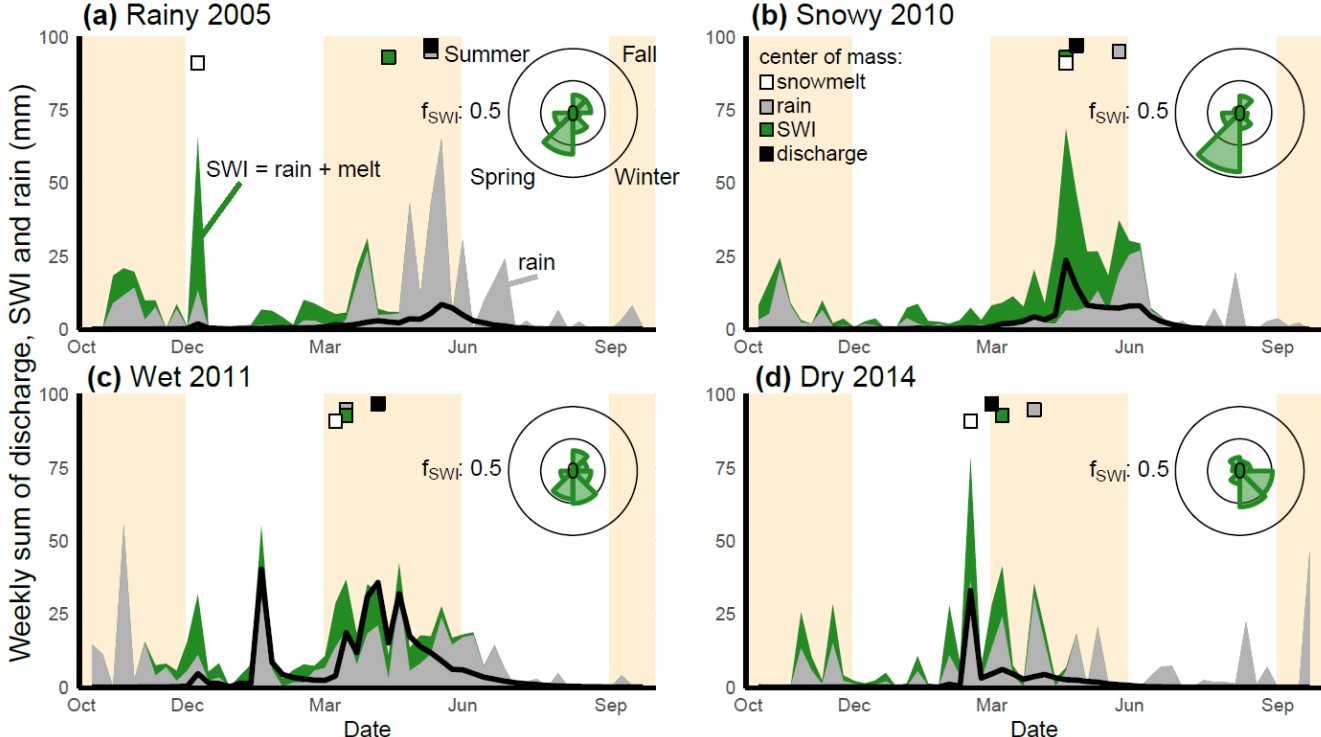

Fig. 4 Weekly sums of surface water inputs (SWI, summation of rainfall and snowmelt, green polygons, mm), rainfall
(grey polygons, mm) and specific discharge (black line graph, mm) for (a) rainy 2005, (b) snowy 2010, (c) wet 2011 and
(d) dry 2014. Background panels are coloured according to the different seasons (fall, winter, spring, summer, fall).
The polar diagram insets indicate the fraction of SWI ($f_{SWI}$) in each season. Squares at the top of each panel indicate
the annual center of mass for snowmelt (white), rainfall (grey), SWI (green) and discharge (black).


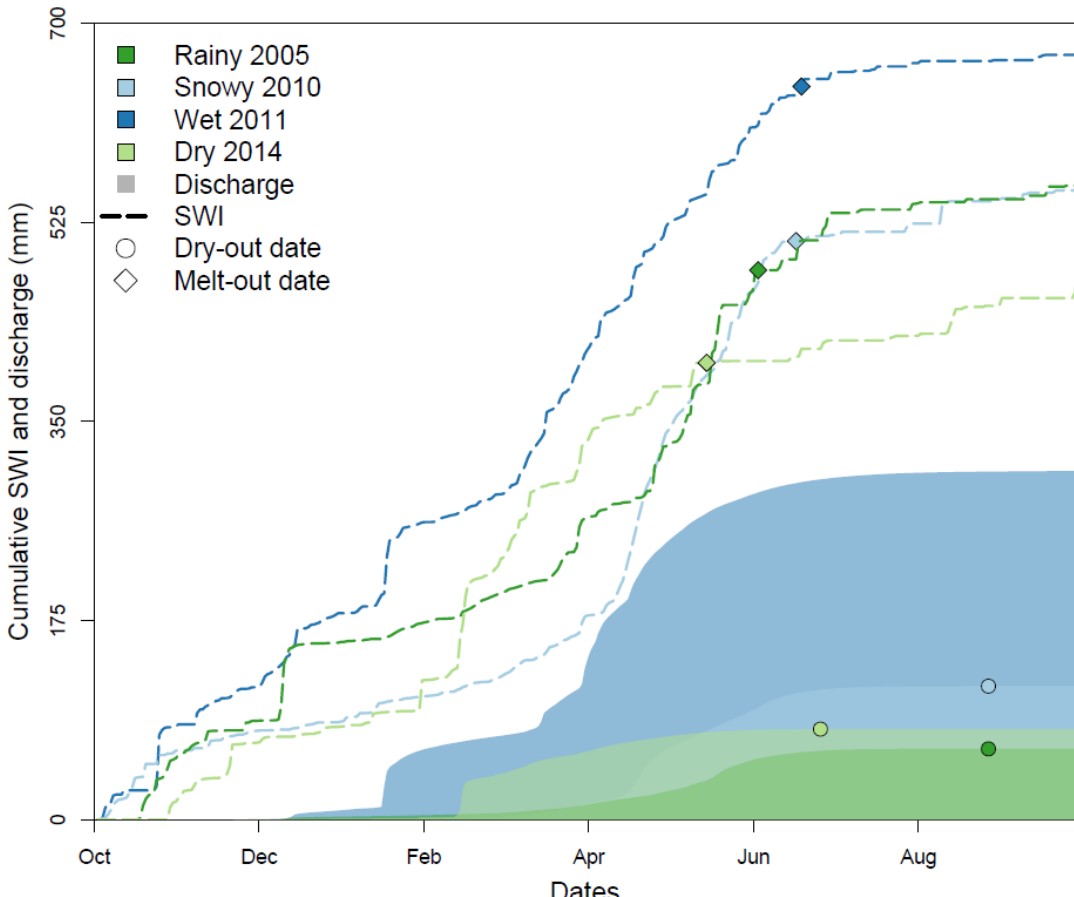

**Fig. 5 Cumulative surface water inputs (SWI, dashed lines, mm) and discharge (coloured polygons, mm) for each of**
**the water years (dark green = rainy 2005, light blue = snowy 2010, dark blue = wet 2011, light green = dry 2014). Circles**
**indicate the day at which the stream ceased to flow at the catchment outlet (dry-out date, please note that the stream**
**did not cease to flow in 2011) and diamonds indicate the day at which all snow had melted from the catchment (melt-**
**out date).**

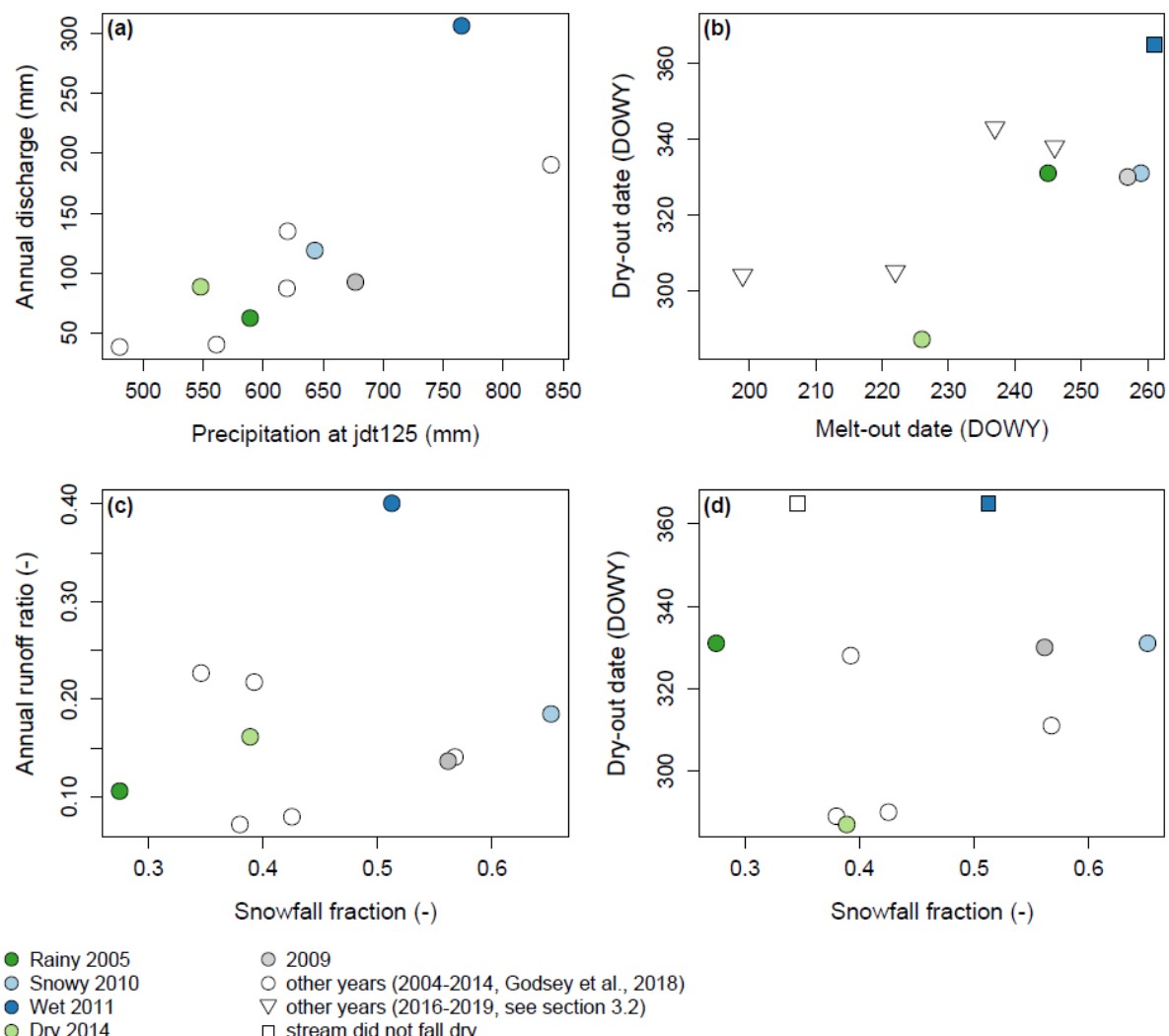

**Fig. 6 Scatter plots of (a) annual discharge at the catchment outlet (mm) and annual precipitation at the lowest**
**precipitation gauge (jdt125, mm; see Supplemental Fig. S14 for a comparison with simulated mean catchment**
**precipitation), (b) the day that surface flow in the stream ceased (dry-out date, day of water year (DOWY)) and the**
**day on which all snow had melted (melt-out date, DOWY), (c) annual runoff ratio (annual discharge/annual**
**precipitation at jdt125) and the annual snowfall fraction (-), and (d) the stream dry-out date and the annual snowfall**
**fraction. Years in which the stream did not dry out are projected to the last day of the hydrological year. $R^2$ and p-**
**values for linear regressions between the variables in each panel are: (a) $r^2$=0.83, p-value=0.001, (b) $r^2$=0.74, p-**
**value=0.023, (c) $r^2$=0.23, p-value=0.524, (d) $r^2$=0.12, p-value=0.730.**

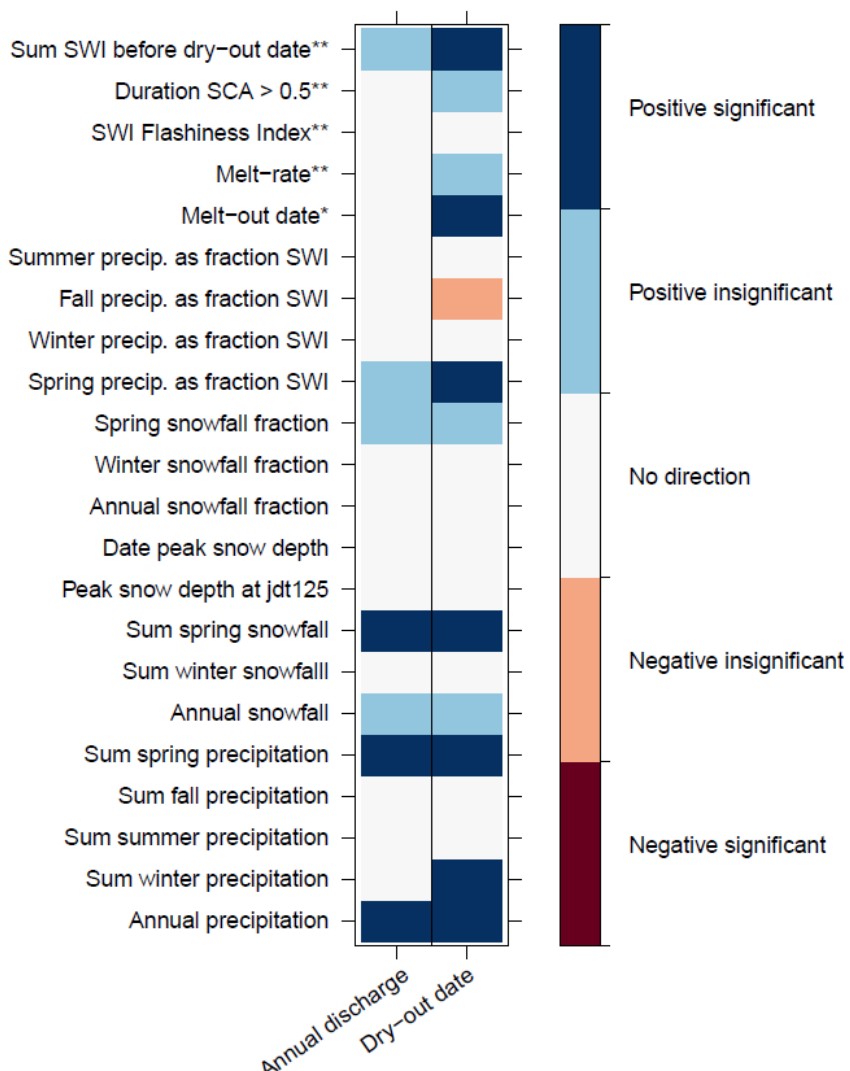

**Fig. 7: Heatplot showing Pearson correlation coefficients (α=0.1) for comparisons between annual discharge, the stream dry-out date and precipitation and snowpack metrics. Significant correlations are marked in dark red (negative) and dark blue (positive), whereas insignificant correlations are marked in light blue (positive) or light red (negative) and correlations without a direction are marked in white (r < 0.3). For most metrics, the comparison is based on the 2004-2014 data record (n=11 years). The comparison with the melt-out date (marked with one asterisk) is based on the simulated years (n=5) and the years for which satellite imagery was available (2016-2019, n=4; which totals to n=9). For the SWI flashiness index, the melt rate, and the number of days when at least half the catchment was snow-covered and the sum of SWI before the dry-out date (marked with two asterisks), we used only the years that were simulated (n=5). Scatter plots of all significant correlations can be found in Supplemental Fig. S9.**