# Peer review of "Effects of spatial and temporal variability in surface water inputs on streamflow generation and cessation in the rain- snow transition zone"

_Hydrology and Earth System Sciences, 2021_

## Author Comment (AC1)

**Dear reviewer,**

We thank you for your time spent reviewing our article, and for the comprehensive and constructive comments. Please find our responses (in blue) to your specific comments (in black) below.

**Emphasizing novelty of the work**

In my opinion, the novelty of the study should be better described. I agree that the focus on the rain-snow transition zone is important and particularly novel, but I would encourage authors to better highlight research gaps and how the study goes beyond to what has been done in the past. Therefore, some additional justification can be added to introduction section (e.g., after research questions).

We are thankful for your comment on clarifying the novelty of our work, and will certainly address this in a revised version of the manuscript. We anticipate elaborating on the following two aspects in a short section after the research questions:

1) In contrast to the majority of snow research, this work is conducted in the rain-snow transition zone – a zone that currently covers a significant area of the mountainous western US and might yield insights in the future functioning of areas that are currently seasonally snow-covered.

2) In contrast to other work that often summarizes daily to seasonal responses at watershed/landscape scales, we quantified surface water inputs (SWI) at a high temporal (hourly) and spatial resolution (10-m). These high-resolution SWI estimates allowed us to investigate:

- The spatial variability in snow depths and SWE in a catchment that has a largely intermittent snow cover. In particular, this revealed the importance of snow drifts even at the rain-snow transition zone.
- The extent to which the temporal distribution of SWI affects stream discharge and stream drying, and how that compares to annual metrics such as snowfall fractions or total precipitation, which are frequently used in larger scale estimations.

In addition, while SWE is frequently used as a summarizing variable for winter precipitation when comparing precipitation to stream discharge, SWI is more directly related to the timing and amount of water resources, and might therefore be an important variable to model in future work addressing similar questions.

If you have further points in mind that can be emphasized, we welcome your reply.

**SWI and model description**

Although authors used frequently applied iSnobal/AWSM model, which is well enough described in the literature, it would be good to provide the reader with more specific information about generating snowmelt runoff, which is specifically important for SWI calculation. For example, how does the model calculate snowmelt? For rain-on-snow situations, is the rainwater directly added to SWI at the specific time or is it temporarily stored and delayed in the snowpack? Does model account for refreezing? Does model consider sublimation from snowpack and canopy interception? These details are not fully described in the current manuscript, but I think they might help the reader with better understanding of how the SWI were calculated.

We recognize that our description of SWI was rather brief, and that adding more information on how SWI is generated in the model will be helpful for the reader without referring to other articles. We will add this information to the revised manuscript and will be sure to cover the topics mentioned in your comment. Below you find the answers to the questions posed in your comment, which hopefully gives some insight into what a description might include in the revised manuscript.

1) How does the model calculate snowmelt?

iSnobal solves each component of the energy balance equation for each model time step using the best available estimations of forcing inputs. Melt occurs in a pixel when accumulated input energy is greater than the energy deficit (i.e. cold content) of the snowpack.

2) Is the rainwater directly added to SWI at the specific time or is it temporarily stored and delayed in the snowpack?

Rainfall is only directly accounted as SWI when it occurs over bare ground. During rain-on-snow events, it is included in the energy and water balances. If the energy deficit in the snowpack is exceeded, this results in snowmelt and thus, SWI, but it will be counted as "SWI from snowmelt", because it did not occur over a bare ground pixel.

3) Does the model account for refreezing?

Yes. In order for runoff (i.e., SWI) to occur, the accumulated melt and liquid water content from the previous hour must exceed a defined threshold. Otherwise, the sum of current hour melt and previous hour liquid water content will be carried over into the next hour. If that hour's input energy conditions are negative, that liquid mass is refrozen into the column.

4) Does [the] model consider sublimation from snowpack and canopy interception? Yes and no. Sublimation from the snow surface is computed and combined with the other mass loss processes (evaporation and condensation) as a model output term. As for canopy interception, iSnobal only predicts snowpack on the ground and is not a comprehensive land surface model. Interception must be handled a priori when developing the model forcing input. Although not accounting for the latter introduces

some uncertainty, we expect this to be small with the vegetation types in Johnston Draw.

**Single lidar observation and poor model performance in WY2011**

L 197: As authors correctly stated, the use of only one lidar survey to describe the snowpack spatial distribution for all study years brings some uncertainty. I see the point that the topography is the main control of snowpack variability. Nevertheless, the meteorological controls might be important as well, such as wind speed and direction influencing snow redistribution and accumulation on leeward sites of slopes. What is the prevailing wind direction? And was it same for all years during snowfall events (and thus likely causing same snowpack distribution)? I would like to see a bit more discussion related to the topic.

We investigated these points further (see below), and will add this discussion to the text of the revised manuscript.

Table 2, Fig. S4: The model performance for north-facing stations and in the "Upper region" (Table 2) in the water year 2011 is relatively poor when comparing simulated and observed SWE values. In addition, even for one single station, simulations for some years are well enough, while this is not the case for another years (e.g., jdt1 and jdt4). Is there any explanation for both temporal and spatial differences in model performance? How confident are observed SWE data for individual stations?

We agree that using a single lidar survey observation raises the question if this observation is truly representative of the snowpack distribution during all years. To address this concern, and to further investigate differences in model performance between years, we now calculated the average wind directions, wind speeds and snow densities for all events during which the snowfall fraction was higher than 0.2 (i.e., 20%) in each year. We used the observational wind speeds and directions from wind-exposed station jdt124, which is located close to the top of the catchment, because this station is most representative of wind along the ridge/scour zone. We computed the averages by considering the impact of each event equally, but also by calculating a weighted average based on the amount of precipitation and snowfall fraction (Table R1.1). We also included a summary of the wind speeds and directions during the 2004-2014 data record for the entire period and during storms and storm-free periods (Figure R1.1).

We suspect that the combination of a higher snow density (stronger cohesion of snow particles) and lower wind speed (less energy for transport) in 2011 compared to 2009 might have led to less wind-redistribution of snow in that year. This might explain the strong underpredictions of snow depths at north-facing and high-elevation sites (jdt3, jdt4, jdt5 and jdt124b). This effect would have been exacerbated compared to 2014 because snowpacks in 2014 were much shallower. Since NSE values are based on squared errors, the divergence between the simulated and observed higher snow depths in 2011 would have resulted in a relatively lower performance in that year.

**Table R1.1**: Average and weighted average of snow densities (Density, simulated) and wind speed ($W_s$, observed) and direction ($W_d$, observed) during events with an average snowfall fraction of more than 0.2 for each water year.

| WY | Average | | | Weighted average | | |
|---|---|---|---|---|---|---|
| | Density (kg m$^{-2}$) | $W_s$ (m s$^{-1}$) | $W_d$ (°) | Density (kg m$^{-2}$) | $W_s$ (m s$^{-1}$) | $W_d$ (°) |
| 2005 | 124 | 4.1 | 187 | 162 | 4.8 | 202 |
| 2009 | 102 | 5.6 | 245 | 102 | 6.5 | 252 |
| 2010 | 24 | 6.6 | 269 | 45 | 8.1 | 272 |
| 2011 | 117 | 5.5 | 232 | 122 | 5.7 | 246 |
| 2014 | 115 | 6.0 | 258 | 126 | 6.1 | 266 |

[Figure]

**Figure R1.1:** Wind roses for stations jd125 (near the catchment outlet), 124 (near the ridge) and jdt3b (a mid-elevation station on the south-facing slope), compiled with data from 2004-2014 (Godsey et al., 2018). The left-hand column includes all measurements, whereas the center and right-hand column only include measurements during storms and storm-free periods, respectively. White indicates higher (> 10 m s$^{-1}$), orange intermediate (5-10 m s$^{-1}$) and brown lower (0-5 m s$^{-1}$) wind speeds.

The snow density, wind speed and wind direction values in 2005 suggest that perhaps, the 2005 simulations might diverge the most from the lidar-derived snow observation in 2009. However, these potential differences will have gone unnoticed because there was only location that recorded snow depths in that year, for which the model performed relatively well (NSE: 0.83). We think that these additional values give more insight into differences between years, and we will describe these in the revised manuscript. The varying performance for simulations at lower stations (jdt125 and jdt1) remains unsolved. We suspect that this might be related to inaccuracies in calculating the phase of precipitation, which would most strongly affect lower elevations at which the phase shifts more often from rain to snow.

We would like to emphasize that despite the low performances for some years and locations, the normalized snow depths were largely acceptable (only five out of 40 year/location-combinations had an $NSE_{norm}$ value below 0.5), which lends confidence that the simulation of ablation and accumulation processes in the model is reasonable.

Regarding the question of how much certainty we have in SWE observations at individual stations: firstly, because only snow depths are available and not SWE, we know that differences in snow density could introduce mismatches between the observed and simulated depths (explained in the manuscript in L385-388). However, in one year at one station, the predicted snow depths were up to 30 cm lower than the observed snow depths, which clearly exceeds estimated offsets due to snow density. Secondly, differences might be introduced because the footprint of the sensor and the cell-size of the model don't match. As installed, the sonic depth sensors have a footprint of ~1-3 m, whereas the simulated snow depths reflect a 10-m resolution grid cell. This is also mentioned in the manuscript L383-385.

**Yearly snowfall fractions as a metric**

The conclusion that the snowfall fraction is not correlated to annual runoff or day of stream drying is certainly important, but maybe not such surprising. The snowfall fraction does not contain the information about total amount of snowfall, but only its relation to the total amount of precipitation. It means, that a year with high snowfall fraction is not necessarily the year with overall high snowfall. Therefore, it would be maybe interesting to select more characteristics describing the snow conditions in different years (such as amount of snowfall during cold season, annual maximum SWE, amount of snowmelt in spring etc.) to better show whether or not the cold season snowfall could positively influence the stream drying compared to the same amount of rain. Perhaps, the results can be shown in some table (heatmap) of paired correlations between individual characteristics.

We agree that the total amount of snowfall is important in addition to the snowfall fraction and will include this in a revised manuscript. We also appreciate your suggestions for other snowpack and precipitation characteristics to consider in addition to the annual snowfall fraction. Indeed, winter, spring, and the sum of winter and spring snowfall can give a more nuanced analysis of timing of precipitation, and also fits well

with our comparison to timing of SWI. Including several metrics might indeed be efficiently shown in a heat map, and we explored this for several additional metrics in Figure R1.2. For the years in which we have sufficient data available, we will also further investigate the relationships between SWE and the melt-out date and melt-rates, such as in Trujillo and Molotch (2014), and potentially include these as variables to explain stream discharge and/or drying. For now, we calculated the (linear) melt rate based on the amount of time it took to completely melt the snow pack from 40% snow coverage (i.e., SCA = 0.4), and represented snow-coverage as the amount of days that the catchment was covered more than 50% (SCA > 0.5). We also included a flashiness index for SWI (Richards-Baker Flashiness Index; Baker et al., 2004). Scatter plots for significant correlations are shown in Figure R1.3.

Because snowfall fraction at the rain-snow transition varies widely from year to year and is one of the most visible manifestations of hydrological changes in this zone that may affect aboveground storage, we would still like to give extra attention to this metric. Although it may be "unsurprising" to some, others may expect it to directly impact stream discharge and/or stream drying, especially in temporally coarser models (e.g., Berghuijs et al., 2014; an analysis that relies on the Budyko water balance framework) or long-term analyses (e.g., Irannezhad et al., 2014).

[Figure]

**Figure R1.2:** Heatplot showing Pearson correlation coefficients (α=0.1) for comparisons between total discharge, the stream dry-out date and precipitation and snowpack metrics. Significant correlations are marked in dark red (negative) and dark blue (positive), whereas insignificant correlations and correlations without a direction (-0.3<$R^2$<0.3) are marked in light blue (positive), light red (negative) and white, respectively. For most metrics the comparison is based on the 2004-2014 data record (n=11 years). The comparison with the melt-out date (marked with one asterisk) is based on the simulated years (n=5) and the years for which satellite imagery was available (2016-2019, n=4), and for the SWI flashiness index, the melt rate and the number of days when at least half the catchment was snow-covered we used only the years that were simulated (n=5, marked with two asterisks).

[Figure]

**Figure R1.3:** Scatter plots of statistically significant comparisons between precipitation and snowpack metrics and total discharge (blue circles) and stream drying (orange circles). Pearson correlation coefficients ($R^2$) are given at the top right of each panel, with the corresponding p-value in brackets.

**Memory effect**

L 286-288: This part would maybe deserve a bit more attention since it touches the important issue of catchment storage and its "memory effect". I found this partial analysis interesting (despite the fact that results did not confirm an effect of "previous water year precipitation"). Therefore, I suggest some extension of the related text.

We appreciate this suggestion and will include more information about the (lack of) memory effect we found in Johnston Draw. Because the stream dries at the outlet in 16 of the 18 years from 2003-2020, it may be difficult to detect any effect of the previous water year precipitation from surface flow data alone. Unfortunately we do not have any groundwater level data to further investigate if and how the memory effect is reflected in subsurface water storage. The frequent stream drying and high potential evaporation rates in this semi-arid, high desert system do suggest that the water that is accessible to plants will be used in the growing season, reducing any memory effect from the shallow, 'active' subsurface storage.

**Proper SWI accounting to compare with drying**

L 297-300: For day of stream drying, would it make more sense to account for sum of SWI preceding the day of stream drying instead of annual sum of SWI?

Thank you for this suggestion. Indeed, any SWI that occurs after the stream dries out cannot have any effect on the date of stream drying. In this catchment, however, the dry summers usually result in very little additional SWI (2.0%, 0.2%, 1.7%, and 0% of annual SWI for 2005, 2009, 2010 and 2011, respectively), meaning the impact of having used annual SWI during these years did not lead to different conclusions. In 2014, the difference was larger (16.5% of annual SWI occurred after the stream dried), which is partly because the dry-out date of the stream was ~1.5 month earlier than in the other years (13 July vs. 25-26 August). We will adapt the metric in the revised manuscript.

**Figures that better illustrate main findings**

Although, I found the reasoning presented in results and discussion sections correct, the supporting illustrations are, in my opinion, less informative and I am not sure whether they fully support all the results and interpretation. For example, one of the main conclusions is that temporal distribution of SWI is more important than its total amount. While I agree with that, it is difficult to me to clearly see this in figures which mostly shows only time series (Figs. 4 and Fig.5). I do not have any clear suggestion how to make figures more informative and supporting the results, but I would encourage authors to reconsider their illustrations and perhaps add another figure which would better show how the timing of SWI influence the runoff response.

Thank you for this remark. It is for us very important that we capture our main findings in the figures and appreciate your comment to make us aware of this. We will have another look at the figures and brainstorm about how to better represent our finding regarding the timing of SWI and runoff response. Perhaps, a combination with the heat plot suggested earlier (see Figure R1.2 for a preliminary version) might be a helpful way to visualize this.

**Technical corrections**

L 116: The decrease in streamflow should be expressed in mm/decade to be comparable with other characteristics.

Thank you for pointing this out. We now divided the trend (-0.75 * $10^6$ $m^3$/decade) by the surface area (54.44 $km^2$) and will include the new trend as -13.8 mm/decade in the text of the revised manuscript.

L 138: "stage height-discharge relationship". Maybe more common term "rating curve" would be better.

We don't hold any preference between the two terms since we are familiar with both, but will adapt it to rating curve in the revised manuscript to avoid confusion with any future readers.

L 193: "Trujillo et al. (2019, manuscript in preparation)". As it seems from references, this paper has been already published.

The Trujillo et al., (2019) reference refers to an AGU abstract, which has been presented at the 2019 AGU Fall Meeting. The corresponding manuscript is still in preparation. We will add "AGU Fall Meeting" to the bibliography so that this is clear. Also, there should have been a semicolon after 2019. The reference now reads "2019; (in preparation)".

Fig. 6a: The annual discharge is related to the precipitation at jdt125 climate station. Why not to show catchment mean precipitation instead? If I understood correctly, the model interpolates stational data to a catchment scale using some kind of elevation dependency. Therefore, to show catchment precipitation in Fig. 6a makes more sense to me to make it better comparable to catchment runoff.

We used precipitation at the climate station rather than simulated precipitation so that we could include additional years in the dataset (2004-2014) without having to run the model for the additional years. Catchment-average precipitation for the years that we did model was linearly related ($R^2$: 0.93) to the precipitation at jdt125 (Figure R1.4). That precipitation at this lowest elevation station is slightly lower than the simulated catchment-average precipitation, based on four stations, is not surprising, since precipitation increases with elevation. All in all, the strong correlation indicates that using precipitation at this station is not expected to lead to a different interpretation. We will include Figure R1.4 (below) in the supplementary material, and refer to it in the text and/or caption of Figure 6 in the revised manuscript.

[Figure]

**Figure R1.4**: Precipitation at jdt125 (the low elevation precipitation gauge) versus the simulated mean catchment precipitation for the years that were modeled.

Fig. 6b: What the triangles represent? Maybe, there is a mistake in the figure as they represent "other years", but different symbol is used in the legend.

This was a mistake in the legend. The diamonds in the legend (other years, 2016-2019) should have been reversed triangles. We will adapt this in the revised manuscript, and will check the legends and symbols in all figures before resubmitting.

**On behalf of all authors,**

**Leonie Kiewiet**

**References**

Baker, D. B., Richards, R. P., Loftus, T. T., and Kramer, J. W.: A NEW FLASHINESS INDEX: CHARACTERISTICS AND APPLICATIONS TO MIDWESTERN RIVERS AND STREAMS, J Am Water Resources Assoc, 40, 503–522, https://doi.org/10.1111/j.1752-1688.2004.tb01046.x, 2004.

Berghuijs, W. R., Woods, R. A., and Hrachowitz, M.: A precipitation shift from snow towards rain leads to a decrease in streamflow, Nature Clim. Change, 4, 583–586, https://doi.org/10.1038/nclimate2246, 2014.

Godsey, S. E., Marks, D., Kormos, P. R., Seyfried, M. S., Enslin, C. L., Winstral, A. H., McNamara, J. P., and Link, T. E.: Eleven years of mountain weather, snow, soil moisture and streamflow data from the rain–snow transition zone – the Johnston Draw catchment, Reynolds Creek Experimental Watershed and Critical Zone Observatory, USA, Earth Syst. Sci. Data, 10, 2018.

Irannezhad, M., Marttila, H., and Kløve, B.: Long-term variations and trends in precipitation in Finland, Int. J. Climatol., 34, 3139–3153, https://doi.org/10.1002/joc.3902, 2014.

Trujillo, E. and Molotch, N. P.: Snowpack regimes of the Western United States, Water Resour. Res., 50, 5611–5623, https://doi.org/10.1002/2013WR014753, 2014.

---

## Author Comment (AC2)

Thank you for taking the time to review our paper and for your constructive comments. We numbered the comments and provide a response to each below in blue.

*1) Main finding and figures*: I agree with reviewer #1 that the text at some places in the results and discussion sections read clear and sound, but the figures do not always support the conclusions or findings. As mentioned by reviewer 1 especially the conclusion on the temporal distribution of SWI is not easy to extract from the figures. Figure 4 and 5 show SWI and discharge, but because the years differ in so many aspects (ratio of rain and snowmelt, timing of SWI, variability of SWI, Q), it is hard to tell which process caused the discharge response from the timeseries, i.e. to see a clear link between temporal distribution and discharge. The text describes these different aspects, but how to generalize these results more? Maybe some measures related to the timing of the center of volume for rainfall and snowmelt, antecedent conditions before spring or number and timing of melt/rainfall events could give some insights. Could also some measure on spatial and temporal distribution be combined? Probably the authors know best how they drew this particular conclusion and could use that to focus on that aspect in the results/figures more explicitly.

We recognize that our figures were more descriptive of the results, and that a clearer presentation of the findings should be included in the figures. In our response to reviewer 1, we suggested adding a heatplot (Fig. R1.2), which based on your feedback, we have now updated to also include the fraction of annual SWI occurring in each season (Fig. R2.1, below). In this new figure, we evaluate how different metrics of the (temporal) distribution of SWI are linked to both annual stream discharge and the day that the stream dried up. We think that different aspects of the temporal distribution of SWI are better represented in this figure. Because the figure highlights statistically significant and insignificant correlations, the reader can more easily connect the discharge variables with SWI magnitude and timing variables and determine that the snowfall fractions do not correlate significantly with discharge or dry-out date.

[Figure]

**Figure R2.1:** Heatplot showing Pearson correlation coefficients (α=0.1) for comparisons between annual discharge, the stream dry-out date and precipitation and snowpack metrics. Significant correlations are marked in dark red (negative) and dark blue (positive), whereas insignificant correlations (-0.3<R²<0.3) are marked in light blue (positive) or light red (negative) and correlations without a direction are marked in white. For most metrics, the comparison is based on the 2004-2014 data record (n=11 years). The comparison with the melt-out date (marked with one asterisk) is based on the simulated years (n=5) and the years for which satellite imagery was available (2016-2019, n=4; which totals to n=9). For the SWI flashiness index, the melt rate, and the number of days when at least half the catchment was snow-covered (marked with two asterisks), we used only the years that were simulated (n=5).

*2) Study setup*: While going through the manuscript I was wondering why only four years were selected. Because of the many processes that influence the discharge signal, a larger sample of years may have provided stronger evidence how processes relate, i.e. avoid that for example the dry year that was analyzed had many rainfall events. From the data description it is a bit unclear to me what the maximum possible amount of years could have been for analyzing. The decision may have to do with the runtime of the model? At least I would expect some description how the selected years deviate from the mean hydro-climatology of the

catchment. Maybe the discussion/limitations section could elaborate on the selection of the years and the intertwined processes when looking at observations and possibilities for future model experiments, isolating some of these aspects (for which discharge would needed to be simulated as well) – but this last point as the authors see fit.

We selected four years because setting up and running the model was a non-trivial task. Also, we aimed to focus on differences in the distribution of SWI and stream discharge for years that had different snowfall ratios and total water inputs and therefore, we selected strongly contrasting years from the 11 potential years of record (Godsey et al., 2018). We will describe this rationale in the revised manuscript, and highlight how the selected years differed from the long-term average: each year's precipitation, snowfall fraction and air temperature is included in Table R2.1 (see below), which will be included in the revised supplementary material. We now also summarize how the years were different from the long-term average in the methods section of the revised manuscript, and will include the information from Table R2.1 in Table 1 of the manuscript.

We also included a scatter plot of annual snow fraction and annual precipitation (Fig. R2.2, see below), that shows that the years we chose to simulate contrasted with the other years captured in the dataset. Although 2007 was slightly drier than 2014 and 2006 was slightly wetter than 2011, we chose to simulate 2011 and 2014 because additional weather stations had been installed in 2011. Temperature and humidity data from these additional stations increased model accuracy and snow depth data from these locations was used to validate the model outputs. We will also include this figure in the revised supplementary material.

**Table R2.1:** Annual precipitation (P, mm), snowfall fractions (SF, -) and air temperature ($T_a$, °C), as well as % of the mean of the 2004-2014 record (Godsey et al., 2018). Simulated years are printed bold.

| | P (mm) | % P | SF (-) | % SF | $T_a$ (°C) | % $T_a$ |
|---|---|---|---|---|---|---|
| 2004-2014 | 524 | 100 | 0.37 | 100 | 8.2 | 100 |
| 2004 | 470 | 90 | 0.49 | 132 | 8.4 | 103 |
| **2005** | **543** | **104** | **0.23** | **63** | **8.2** | **100** |
| 2006 | 714 | 136 | 0.29 | 78 | 8.4 | 103 |
| 2007 | 402 | 77 | 0.31 | 83 | 9.3 | 113 |
| 2008 | 465 | 89 | 0.45 | 123 | 7.4 | 91 |
| **2009** | **549** | **105** | **0.49** | **132** | **8.0** | **98** |
| **2010** | **531** | **101** | **0.57** | **155** | **6.6** | **81** |
| **2011** | **693** | **132** | **0.41** | **111** | **7.4** | **91** |
| 2012 | 494 | 94 | 0.24 | 64 | 8.6 | 105 |
| 2013 | 456 | 87 | 0.26 | 72 | 8.6 | 105 |
| **2014** | **450** | **86** | **0.30** | **82** | **8.6** | **105** |

[Figure]

**Figure R2.2:** Scatterplot of the annual precipitation and snowfall fraction of precipitation at weather station jd125, which is located close to the catchment outlet. Simulated years are shown in blue, other years are shown in black.

*3) Argumentation in introduction*: Partly related to the comment of reviewer 1 on a better description of the novelty in the introduction, I think that the line of thoughts for this study and the research gap can be better described. In my opinion, the introduction mixes 1) changes in snowmelt generated streamflow, 2) differences between catchments seasonally snow covered and in the rain-to-snow transition zone, 3) rain-to-snow zones as a space-for-time substitution of catchments that are now seasonally snow covered and 4) changes that have occurred in the rain-to-snow transition zone and may occur in the future. Although all of these aspects may be important to put the study into context, I would suggest to clearly identify the research gap (how do yearly variations in rainfall and snowmelt influence discharge, relation with snowfall fraction not yet clear, rain-to-snow zone suitable to analyze 'extremes', i.e. snowy and rainy) and explain the implications for future changes and

relations to observed changes in different type of catchments in a more structured way.

Thank you for this comment. We agree that it is important to clearly describe the research gaps and rationale behind the study, and will revise the introduction to emphasize the following points:

1) In contrast to the majority of snow research, this work is conducted in the rain-snow transition zone – a zone that currently covers a significant area of the mountainous western US and might yield insights in the future functioning of areas that are currently seasonally snow-covered.

2) In contrast to other work that often summarizes daily to seasonal responses at watershed/landscape scales, we quantified surface water inputs (SWI) at a high temporal (hourly) and spatial resolution (10-m). These high-resolution SWI estimates allowed us to investigate:

- The spatial variability in snow depths and SWE in a catchment that has a largely intermittent snow cover. In particular, this revealed the importance of snow drifts, even at the rain-snow transition zone.
- The extent to which the temporal distribution of SWI affects stream discharge and stream drying, and how that compares to annual metrics such as snowfall fractions or total precipitation, which are frequently used in larger scale estimations.

In addition, while SWE is frequently used as a summarizing variable for winter precipitation when comparing precipitation to stream discharge, SWI is more directly related to the timing and amount of water resources, and might therefore be an important variable to model in future work addressing similar questions.

*4) Methods and data description*: Here I missed some details regarding the available data, the model and the choice of years. As indicated above, it is not mentioned how the four climatologically different years were selected. I was also a bit confused by the numbers in table 1, how come that in a rainy year, the SWIsnow is higher than in a snowy year? Are numbers switched here? And without knowing the range of snowfall fractions over a longer time period it is difficult to interpret the values of the different years. It would also be helpful to explain the reasoning and possible hypotheses of selecting rainy and snowy years and wet and dry years. Could temperatures also be given for the years? Regarding the data and model, what is needed as input for the model? And which of the stations do have this data available for which time period.

We recognize that the description of the data, model and selection of years was rather short. We will add more information on the functioning of the model (e.g., how the model calculates snowmelt, how rainwater is handled during rain-on-snow

events, how refreezing is represented and how sublimation is considered), as described in the response to reviewer 1, as well as information on the required model input and how that overlaps with availability of data for the different years. We will also describe the selection of the simulated years better with the information described above at comment 2, summarized in Figure R2.2 and Table R2.1. This should provide context for the years that we chose to simulate as well as on the dataset as a whole.

Thank you for pointing out the mistake in Table 1. It should have been 412 mm and 243 mm for SWI from rain in 2005 and 2010 respectively, and 146 mm and 310 mm for SWI from snowmelt in 2005 and 2010, respectively. We will check all values in this table (and the other tables) before submitting the revised manuscript.

*Minor and technical corrections:*

5) Title + abstract: 'Snowfall fractions' – since you only clarify in the introduction, maybe another term could be used here, e.g. ratio of snowfall to precipitation. Regarding the title, maybe it needs to be adjusted depending on the changes, e.g. temporal distribution and total input? Or specify what is meant with temporal distribution. Stream discharge – Annual (stream) discharge.

We recognize how the title and abstract could be more explicit, and are open to adjusting these after we have made all the changes in the manuscript.

L13 '..spatial and temporal distribution of precipitation' – add phase of precipitation?

Yes indeed, precipitation phase might also impact stream discharge. We will add this to the abstract as "spatial and temporal distribution of precipitation and precipitation phase"

L68 which catchments?

We think that our findings will be most applicable to other small (<10 km$^2$), semi-arid, mid-elevation, mid-latitude catchments, and will include that as specification. We suggest these catchment characteristics because we think that 1. similarly sized catchments are more likely to have a similar potential for water storage on the surface and in the subsurface, 2. semi-arid, mid-latitude catchments are likely to have a similar vegetation cover 3. mid-latitude, mid-elevation catchments are likely to have a temporal distribution of water inputs that is similar to that in Johnston Draw.

L71-72: on an annual time-scale is this so different, apart from the effects of snow redistribution? Is this something interesting to show for you analyses, i.e. spatial distribution of rainfall and spatial distribution of snowmelt?

Apart from the snow redistribution, distribution of rainfall and snowfall might be quite similar across the catchment, and in both cases increase slightly with elevation. However, snow redistribution causes surface water inputs to differ across the catchment, even at the rain-snow transition zone. We will adapt the text of this paragraph to make sure that is clear for the reader.

When analyzing the data, we visualized SWI inputs from rainfall and snowmelt separately, but refrained from including these figures in the manuscript. We decided this because rainfall amounts were interpolated following a linear orographic gradient derived from precipitation at the gauges at the upper and lower end of the catchment. Because of this, rainfall distribution across the catchment did not reflect any small-scale variations in the spatial distribution of rainfall, other than that caused by elevation. The effects of wind-redistribution of snow were implicitly included in the spatial distribution of snowfall, which was based on the lidar snow depths, and thus included more fine-scale spatial variation. Hence, a direct comparison with the orographic precipitation gradient might overrepresent the differences. Investigating how the spatial distribution of non-redistributed snowfall and rainfall might differ could be achieved by simulating the snowpack with and without wind re-distribution, but we think this is outside of the scope of the manuscript presented here.

L94: 'However' – where does this refer to?

This was meant to refer to the difference in catchment wetness that might exist between rainfall versus snowmelt dominated catchments. We recognize how the writing here might have been confusing, and will remove the word 'however' so that the sentence now reads: "Rain and snowmelt inputs might result in similar runoff ratios (discharge/SWI) as long as the overall catchment wetness is similar or if the catchment is wet at key locations for water transport."

L116-117: did increased ET played a role here?

While the long-term analysis of Nayak et al. (2010) does not comment on increased evaporation or transpiration, Seyfried et al. (2011) states that evapotranspiration is most sensitive to increases in PET (implied by increases in air temperature) during ~4-5 weeks each year in which the plants have developed leaves and sufficient water is available in the soil. Before that time, plants use little water, and after that time, the system is strongly water-limited. Hence, although increased plant water use might be important in some systems, we suspect that it might not strongly affect stream discharge in this region. We will include a small summary of this information in section 2 of the revised manuscript.

L195-196: 'this uncertainty…. Patterns' – double with few sentences above

This part of the sentence repeats itself because we wanted to highlight the connection between the intra-annual consistency in snowpack patterns introduced above, and the uncertainty related to using the snow-on lidar from only one year, discussed here. To avoid the exact repetition of words, we will change this part of the sentence in L195 to something like: "… might have induced some uncertainty, but this uncertainty is likely to be small given the consistent spatial snow distribution, and was verified in this catchment …"

Section 3.5 How do catchment precipitation and discharge compare? Are there estimations for ET?

Precipitation and discharge are significantly correlated ($R^2$= 0.6, p-value = 0.005), which is shown in Fig. 6a of the original manuscript. There are no estimations for ET in this catchment. However, there are estimations for a nearby catchment at slightly higher elevation (1930 m) and that receives less precipitation (Upper Sheep Creek; Flerchinger et al., 1998; MAP: 479 mm versus 609 mm for Johnston Draw). Their measurements showed that evapotranspiration depends significantly on precipitation inputs, and amounted to 58% of annual precipitation for a wet year (703 mm precipitation) and 95% for a 'normal' year (482 mm of precipitation). Although estimating ET for the years and catchment presented here is beyond the scope of this effort, we will include the information about Upper Sheep Creek in the discussion.

L223 'this pattern was masked by the effects of other processes' – what is meant here? In general in the results section it would be helpful to indicate better when observations or when simulations are described.

Here we mean the snow redistribution processes, and we further detail measurements in which this process can be observed in L223-228. We agree that being specific is helpful, and we will change the text to "…the snowpack distribution was also affected by wind-driven redistribution of snow. For instance, the snow depths at jdt2 …."

L236-237 'differential melt-out patterns' – what was compared for that?

We compared the simulated persistence of the snowpack with the persistence of the snow-covered area from the satellite imagery. This comparison showed that the areas that were simulated to be snow-covered longer were also snow-covered longer in the satellite imagery. We recognize that we did not explain that very clearly in the manuscript and will add this to section 4.2 of the revised manuscript.

L267 'As a result, average daily SWI rates were higher' – as a result of what?

We meant to say here that average daily SWI rates were higher as a result of higher snowfall and lower rainfall inputs. We will adapt this reference in the revised manuscript so that this is clear.

L274 'whereas roughly 30% of SWI....' – are delays taken into account, or is meant here the comparsion between SWI from month x to month y and discharge from month x to month y? Are the events where Q is higher than SWI also of interest?

This refers to the comparison of SWI in period x and discharge in period x, and no delays are taken into account. We will clarify this in the methods section of the revised the manuscript. Events where Q is higher than SWI are definitely of interest, but these are not further explored in this manuscript because we did not investigate discharge generation during individual events.

L279 Have you tried plotting % of SWI translated into discharge against temperature (annual, or during growing season?)

We did not do this, but find it an interesting suggestion! Plotting runoff efficiency as discharge/precipitation vs. mean air temperature shows that they are weakly and not significantly correlated ($R^2$= -0.43, p-value=0.217; Fig. R2.3). Perhaps, this corroborates that evapotranspiration is water-limited in this system rather than energy-limited. We will allude to this small additional analysis in the revised manuscript.

[Figure]

**Figure R2.3:** Annual air temperature (°C) versus runoff efficiency (discharge/precipitation, mm mm$^{-1}$).

We also investigated if the runoff ratio was related to precipitation or SWI on the seasonal scale (i.e., spring, summer, fall winter). We found that Q/P was weakly correlated to air temperatures during each season, with the summer period yielding the highest correlation ($R^2$=-0.54, p-value=0.08). We assume that temporal offsets between snowfall and snowmelt might have led to low correlations in spring and winter. We found strong but insignificant relationships when calculating the efficiency as Q/SWI (up to $R^2$=-0.72, p-value=0.169, also during the summer season), and suspect that the insignificance of this relationship is likely due to the low number of observations (n=5). Together, these results suggest that temperature likely influences runoff efficiency in the warmer season, but has little effect in the cooler season.

Section 5 – the subsections have no numbering

Thank you. We will number the subsections in the discussion in the revised manuscript.

L357 'This highlights the importance of the temporal distribution of SWI' – also the importance of total water input?

Definitely! Although this was not emphasized in the initial version of the manuscript, we agree that this can be added as a conclusion. We will support this conclusion with the additional heat plot (Figure R2.1) and emphasize this finding in the text.

L360 'events' – throughout the manuscript when using 'event' please check if it is clear why event is meant? Precipitation, rainfall, snowmelt, discharge?

Thanks for pointing this out. We meant 'event' as 'rainfall or snowmelt event' (i.e., an event related to SWI). We now specify the type of event at each of the eight occurrences in the manuscript (e.g., it is written explicitly in the text as precipitation event, rain-on-snow event, snowmelt event…).

L369 'catchment' – sub-catchment?

Yes, 'Treeline' refers to a sub-catchment rather than the entire Dry Creek catchment. We will update this in the revised manuscript.

Discussion on simulated snow depts – could it be extended with a description of the reasons for varying performance for individual years and maybe a hypothesis how such 'bad' simulated years potentially could have influenced the results?

Thank you for this suggestion. We think that a summary of the extensive discussion in our reply to reviewer 1 would be a good addition to the current discussion on simulated snow depths, and aim to include that in the revised manuscript.

In short,

- weighted-average wind directions were similar between most years (246-272° for 2009, 2010, 2011 and 2014), but differed slightly in 2005 (202°).
- we suspect that the combination of a higher snow density (stronger cohesion of snow particles) and lower wind speed (less energy for transport) in 2011 compared to 2009 might have led to less wind-redistribution of snow in that year. This effect would have been exacerbated compared to 2014 because snowpacks in 2014 were much shallower. Since NSE values are based on squared errors, the divergence between the simulated and observed higher snow depths in 2011 would have resulted in a relatively lower performance in that year.

- The snow density, wind speed and wind direction values in 2005 suggest that perhaps, the 2005 simulations might diverge the most from the lidar-derived snow observation in 2009. However, these potential differences will have gone unnoticed because there was only location that recorded snow depths in that year, for which the model performed relatively well (NSE: 0.83).

As to how this might have influenced the results: For years in which the actual snow redistribution was less strong than simulated, snow drifts might have been overrepresented, resulting in a later simulated melt-out date of the snowpack. If the snow redistribution was overestimated in some years, but underestimated in others, this might have resulted in either a stronger or weaker relationship between the snowpack melt-out date and the stream dry-out date.

L419 '…, which influences' – should it be, which may influence? As for example one of your conclusions is that the spatial distribution of SWI stays rather stable over time?

Thank you for this suggestion. Indeed, although we might expect that effects on the spatial distribution could be more severe if snow redistribution patterns are also affected, that is not shown in our work. We will adapt the phrase to "may influence".

L428 -429 Could a short explanation/hypothesis be added why Q was much higher in 2010?

We think that a short explanation can be a nice addition to the conclusions, and will change this part of the conclusions as follows "Despite similar annual SWI (553 vs. 557 mm), snowy 2010 had about twice as much stream discharge as rainy 2005. This is likely related to a higher fraction of SWI occurring in spring 2010 (46%) than in spring 2005 (32%)."

We also checked if the fraction of precipitation occurring in spring was related to annual stream discharge or runoff efficiency, and found a statistically significant positive relationship with the stream dry-out date ($R^2$= 0.58 p-value=0.06), and a positive, statistically insignificant relationship for annual discharge; $R^2$=0.43 p-value=0.18). We now include these findings in the heatplot shown in Fig. R2.1.

[Figure]

**Figure R2.4:** Scatter plot of spring precipitation as a fraction of total precipitation and the stream dry-out date ($R^2$= 0.58 p-value=0.06).

Figure 2e – what do the light coloured pixels mean? Was there no snow cover in the simulations while there was around 0.5 in the satellite observations? Because of the comparison of different years?

The light-colored pixels indicate that the time that an area was simulated to be snow-covered was lower than 0.25 (0-0.25). Indeed, that estimation differs from the fraction of time that these areas were snow-covered based on the satellite imagery (0.5). This difference might be due to a small difference in the mean annual air temperature, which was a bit higher in 2009 than in 2019 (8.0°C versus 6.7°C), which could have resulted in faster melt-out. We will highlight this difference in the text of section 4.2.

For all figures it may be good to not only indicate the year but also its characteristic (i.e. snowy, rainy, wet and dry) in the figure itself instead of the legend.

Thank you for this suggestion. In earlier versions of the figures we indeed included the characteristic of each year in the panel titles, and will re-introduce that for the figures in the revised manuscript.

---

## Author Response (AR1)

**Dear Markus,**

Many thanks for your efforts handling our manuscript. In this submission, we provide you and the reviewers with replies to the review comments and a revised manuscript. In the response to the reviewers we included the line numbers where adaptations to the text have been made. Focus points of the revision included (but were not limited to):

- Adaptation of the introduction and emphasis of the novelty of the work
- A new figure to make a stronger link between the conclusions in the text and figures
- Clarification of some features of the model
- Clarification of the choice of simulated years
- A summary of the meteorological characteristics of each year and how these characteristics might have affected the modeling efficiency

We think that the edits helped to improve the manuscript, and look forward to feedback from you and the reviewers.

On behalf of all co-authors,

Leonie Kiewiet

**Reviewer 1**

Dear reviewer,

We thank you for your time spent reviewing our article, and for your comprehensive and constructive comments. Please find our responses (in blue) to your specific comments (in black) below. The revised text and line numbers detailing where to find these adaptations in the revised manuscript are printed in red.

Kind regards on behalf of all co-authors,

Leonie Kiewiet

**1.1 Emphasizing novelty of the work** In my opinion, the novelty of the study should be better described. I agree that the focus on the rain-snow transition zone is important and particularly novel, but I would encourage authors to better highlight research gaps and how the study goes beyond to what has been done in the past. Therefore, some additional justification can be added to introduction section (e.g., after research questions).

We are thankful for your comment on clarifying the novelty of our work. To do this, we restructured the introduction (see detailed answer in response to review comment 2.3) and added a paragraph after research questions to highlight the novel aspects of this work (L123-131).

"Examining natural variation in snowfall fractions in the rain-snow transition zone contrasts with other research on snow-related processes that focus on seasonally-snow covered catchments. While many studies of snowmelt runoff examine seasonal responses at the landscape scale, here we focus on hourly responses at a fine spatial resolution. This allows us to investigate the spatial distribution of the snowpack and snowmelt, as well as the phase of precipitation and the temporal distribution of SWI. Furthermore, while SWE is frequently used as a summarizing variable for winter precipitation when comparing precipitation to stream discharge, SWI is more directly related to the timing and amount of water resources, and might therefore be an important variable to model in future work addressing similar questions. Lastly, we also compare stream discharge to annual metrics of snowfall fraction and total precipitation to provide a finer-scale context for results from larger scale models or estimations that rely on annual metrics.."

**1.2 SWI and model description** Although authors used frequently applied iSnobal/AWSM model, which is well enough described in the literature, it would be good to provide the reader with more specific information about generating snowmelt runoff, which is specifically important for SWI calculation. For example, how does the model calculate snowmelt? For rain-on-snow situations, is the rainwater directly added to SWI at the specific time or is it temporarily stored and delayed in the snowpack? Does model account for refreezing? Does model consider sublimation from snowpack and canopy interception? These details are not fully described in the current manuscript, but I think they might help the reader with better understanding of how the SWI were calculated.

We recognize that our description of SWI was rather brief, and added more information on how SWI is generated in the model. Below you find the answers to the questions posed in the original review comment, which are summarized in paragraph 3.4 of the revised manuscript (L235-251).

"One of the model outputs from iSnobal is 'surface water input' (SWI), which represents snowmelt from the bottom of the snowpack, rain on snow-free ground, or rain percolating through the snowpack. Rainfall is directly counted as SWI when it falls over snow-free ground, and it is included in the energy and water balances when it falls onto the snowpack. To calculate snowmelt, iSnobal solves each component of the energy balance equation for each model time step using the best available estimations of forcing inputs. Melt occurs in a pixel when the accumulated input energy is greater than the energy deficit (i.e. cold content) of the snowpack. If the accumulated energy input is smaller than the energy deficit, the sum of current hour melt and previous hour liquid water content will be carried over into the next hour. If that hour's input energy conditions are negative, the liquid mass is refrozen into the column. Sublimation and evaporation of liquid water from the snow surface and condensation of liquid water onto the snow surface is computed as a model output term, though these quantities were not considered here. Canopy interception must be handled a priori when developing the model forcing input, and it was also not considered here. Although not accounting for the latter introduces some uncertainty, we expect this to be small with the shrub and grass vegetation types in Johnston Draw. Lastly, iSnobal is limited to snow

processes only, which means that SWI 'exits' the modelling domain. In reality, SWI might travel to the stream as surface or subsurface runoff, could be stored in the soil until it evaporates or is transpired, or could recharge deeper groundwater storages. The route that SWI takes depends on the overall catchment wetness as well as the local energy balance (e.g., incoming radiation) and vegetation activity. In this manuscript, we computed SWI for each pixel and time step and assumed that all SWI generated in simulated snow-free pixels was rain and that all SWI generated in simulated snow-covered pixels was snowmelt."

1) How does the model calculate snowmelt?

This is addressed in the main text (L237-240): "To calculate snowmelt, iSnobal solves each component of the energy balance equation for each model time step using the best available estimations of forcing inputs. Melt occurs in a pixel when accumulated input energy is greater than the energy deficit (i.e. cold content) of the snowpack".

2) Is the rainwater directly added to SWI at the specific time or is it temporarily stored and delayed in the snowpack?

This is now specified in L236-237: "Rainfall is only directly accounted as SWI when it occurs over snow-free ground; otherwise it is included in the energy and water balances when it falls onto the snowpack." and (L250-251): "In this manuscript we … assumed that all SWI generated in simulated snow-free pixels was rain and that all SWI generated in simulated snow-covered pixels was snowmelt.".

3) Does the model account for refreezing?

Yes. From revised L240-242: "If the accumulated energy input is smaller than the energy deficit, the sum of current hour melt and previous hour liquid water content will be carried over into the next hour. If that hour's input energy conditions are negative, the liquid mass is refrozen into the column."

4) Does [the] model consider sublimation from snowpack and canopy interception? Yes and no, respectively, as explained more fully in the revised text (L242-246): "Sublimation and evaporation of liquid water from the snow surface and condensation of liquid water onto the snow surface is computed as a model output term, though these quantities were not considered here. Canopy interception must be handled a priori when developing the model forcing input, and it was also not considered here. Although not accounting for the latter introduces some uncertainty, we expect this to be small with the shrub and grass vegetation types in Johnston Draw."

**1.3 Single lidar observation and poor model performance in WY2011** L197: As authors correctly stated, the use of only one lidar survey to describe the snowpack spatial distribution for all study years brings some uncertainty. I see the point that the topography is the main control of snowpack variability. Nevertheless, the meteorological controls might be important as well, such as wind speed and direction influencing snow redistribution and accumulation on leeward sites of slopes. What is the prevailing wind direction? And was it same for all years during snowfall events (and

thus likely causing same snowpack distribution)? I would like to see a bit more discussion related to the topic.

To address the variations in model performance between years, we investigated the impact of wind direction, wind speed and snow density for all events during which the snowfall fraction was higher than 0.2 (i.e., 20%), and added these findings to the manuscript (L450-461, also below). Furthermore, Table R1.1 and Fig. R1.1 (see next two pages of this response) now appear in the revised supplementary material as Table S12 and Figure S5, respectively.

"To investigate why simulations of snow depths were poor for some stations and years, we calculated the average and precipitation-weighted average wind directions, wind speeds and snow densities for all events during which the snowfall fraction was higher than 0.2 (i.e., 20%; see Supplemental Table S12 and Fig. S13) from the station data. Although wind speed and directions were generally consistent (Supplemental Fig. S13), in 2011, the combination of higher snow densities (stronger cohesion of snow particles; 122 kg m-2) and lower wind speeds (less energy for transport; 5.7 m s-1) compared to 2009 (102 kg m-2 and 6.5 m s-1, respectively, precipitation-weighted averages in Table S12) might have led to less wind redistribution of snow in that year and correspondingly resulted in underpredictions of snow depths at north-facing and high-elevation sites in 2011 (jdt3, jdt4, jdt5 and jdt124b). Since NSE values are based on squared errors, the divergence between the simulated and observed snow depths impacted the model performance more severely in 2011 than in years with shallower snowpacks (i.e., 2005 and 2014). The snowpack density, wind speed and wind direction values in 2005 diverged most from 2009, from which the lidar observations were used, but nonetheless had a relatively high performance (NSE: 0.83), possibly because there was data from only one station available for validation."

**-more detailed discussion-**

Table 2, Fig. S4: The model performance for north-facing stations and in the "Upper region" (Table 2) in the water year 2011 is relatively poor when comparing simulated and observed SWE values. In addition, even for one single station, simulations for some years are well enough, while this is not the case for another years (e.g., jdt1 and jdt4). Is there any explanation for both temporal and spatial differences in model performance? How confident are observed SWE data for individual stations?

We agree that using a single lidar survey observation raises the question if this observation is truly representative of the snowpack distribution during all years. As noted above, we calculated the average wind directions, wind speeds from jd124 and snow densities for all events during which the snowfall fraction was higher than 0.2 (i.e., 20%) in each year. We computed the averages by considering the impact of each event equally, but also by calculating a precipitation-weighted average based on the amount of precipitation and snowfall fraction (Table S12). We also included a summary of the wind speeds and directions during the 2004-2014 data record for the entire period and during storms and storm-free periods (Figure S5).

Table S12: Average and weighted average of snow densities (Density, simulated) and wind speed ($W_s$, observed) and direction ($W_d$, observed) during events with an average snowfall fraction of more than 0.2 for each water year.

| WY | Average | | | Weighted average | | |
|---|---|---|---|---|---|---|
| | Density (kg m$^{-2}$) | $W_s$ (m s$^{-1}$) | $W_d$ (°) | Density (kg m$^{-2}$) | $W_s$ (m s$^{-1}$) | $W_d$ (°) |
| 2005 | 124 | 4.1 | 187 | 162 | 4.8 | 202 |
| 2009 | 102 | 5.6 | 245 | 102 | 6.5 | 252 |
| 2010 | 24 | 6.6 | 269 | 45 | 8.1 | 272 |
| 2011 | 117 | 5.5 | 232 | 122 | 5.7 | 246 |
| 2014 | 115 | 6.0 | 258 | 126 | 6.1 | 266 |

[Figure]

Figure S5: Wind roses for stations jd125 (near the catchment outlet), 124 (near the ridge) and jdt3b (a mid-elevation station on the south-facing slope), compiled with data from 2004-2014 (Godsey et al., 2018). The left-hand column includes all measurements, whereas the center and right-hand column only include measurements during storms and storm-free periods, respectively. White indicates higher (> 10 m s$^{-1}$), orange intermediate (5-10 m s$^{-1}$) and brown lower (0-5 m s$^{-1}$) wind speeds.

The snow density, wind speed and wind direction values in 2005 suggest that, perhaps, the 2005 simulations might diverge most from the lidar-derived snow observation in 2009. However, these potential differences will have gone unnoticed because there was only location that recorded snow depths in that year, for which the model performed relatively well (NSE: 0.83), described in L459-461. We think that these additional values give more insight into differences between years, and we described this in the results of the revised manuscript (L450-459). The varying performance for simulations at lower stations (jdt125 and jdt1) remains unsolved. We suspect that this might be related to inaccuracies in calculating the phase of precipitation, which would most strongly affect lower elevations at which the phase shifts more often from rain to snow. This is also included in the revised manuscript (L464-466).

We would like to emphasize that despite the low performances for some years and locations, the normalized snow depths were largely acceptable (only five out of 40 year/location-combinations had an $NSE_{norm}$ value below 0.5), which lends confidence that the simulation of ablation and accumulation processes in the model is reasonable (L308-311).

Regarding the question of how much certainty we have in SWE observations at individual stations: firstly, because only snow depths are available and not SWE, we are aware that differences in snow density could introduce mismatches between the observed and simulated depths (L472-474). However, in one year at one station, the predicted snow depths were up to 30 cm lower than the observed snow depths, which clearly exceeds estimated offsets due to snow density (not explicitly mentioned in the manuscript). Secondly, differences might be introduced because the footprint of the sensor and the cell-size of the model don't match. As installed, the sonic depth sensors have a footprint of ~1-3 m, whereas the simulated snow depths reflect a 10-m resolution grid cell. This is mentioned in the manuscript L468-471.

**1.4 Yearly snowfall fractions as a metric** The conclusion that the snowfall fraction is not correlated to annual runoff or day of stream drying is certainly important, but maybe not such surprising. The snowfall fraction does not contain the information about total amount of snowfall, but only its relation to the total amount of precipitation. It means, that a year with high snowfall fraction is not necessarily the year with overall high snowfall. Therefore, it would be maybe interesting to select more characteristics describing the snow conditions in different years (such as amount of snowfall during cold season, annual maximum SWE, amount of snowmelt in spring etc.) to better show whether or not the cold season snowfall could positively influence the stream drying compared to the same amount of rain. Perhaps, the results can be shown in some table (heatmap) of paired correlations between individual characteristics.

We agree that the total amount of snowfall is important in addition to the snowfall fraction and now included this as 'Total snowfall'. We also appreciate your suggestions for other snowpack and precipitation metrics to consider in addition to the annual snowfall fraction. Indeed, winter and spring snowfall can give a more nuanced analysis of timing of precipitation, and also fits well with our comparison to timing of SWI, hence

we included them as metrics too. We added three other metrics: (1) the (linear) melt rate based on the amount of time it took to completely melt the snow pack from 40% snow coverage (i.e., SCA = 0.4), (2) the number of days that more than 50% of the catchment was covered by snow (SCA > 0.5), and (3) a flashiness index for SWI (Richards-Baker Flashiness Index; Baker et al., 2004).

We now explain the different metrics that are used on L261-271, and show the correlation between total stream discharge, the stream dry-out date, and each of the new and previously used metrics in a heatplot in Fig. 7. Scatter plots for all significant correlations are included in Supplement Figure S9 (pasted below).

[Figure]

**Supplemental Figure S9**: Scatter plots of statistically significant comparisons between precipitation, SWI and snowpack metrics and total discharge (blue circles) and the stream dry-out date (orange circles). Pearson correlation coefficients (r) are given at the top right of each panel, with the corresponding p-value in parentheses.

**1.5 Memory effect** L 286-288: This part would maybe deserve a bit more attention since it touches the important issue of catchment storage and its "memory effect". I found this partial analysis interesting (despite the fact that results did not confirm an effect of "previous water year precipitation"). Therefore, I suggest some extension of the related text.

We appreciate this suggestion and now included more information about the (lack of) memory effect we found in Johnston Draw (L349-357).

"Annual discharge was slightly higher for years that were preceded by a year that received above average annual precipitation (see Supplemental Fig. S8), but the correlation coefficient decreased when including the precipitation totals recorded in the preceding year (e.g., annual discharge vs. precipitation in the same year + 0.5 times precipitation previous year). This indicates that any memory effect is likely to be small in this catchment. Frequent stream drying (16 out of 18 years between 2003 and 2020, data not shown, the stream did not cease flow in 2006 and 2011) and the high potential evaporation rates in this semi-arid, high desert system (evapotranspiration accounts for nearly 90% of precipitation in the nearby Upper Sheep Creek catchment; Flerchinger and Cooley, 2000) also suggest that any water in the shallow, 'active' subsurface storage is likely limited, and that any memory effect, if present, is perhaps constrained to deeper subsurface water storages."

**1.6 Proper SWI accounting to compare with drying** L 297-300: For day of stream drying, would it make more sense to account for sum of SWI preceding the day of stream drying instead of annual sum of SWI?

Thank you for this suggestion. Indeed, any SWI that occurs after the stream dries out cannot have any effect on the date of stream drying. In this catchment, the dry summers usually result in very little additional SWI (2.0%, 0.2%, 1.7%, and 0% of annual SWI for 2005, 2009, 2010 and 2011, respectively), meaning the impact of having used annual SWI during these years did not lead to different conclusions. In 2014, the difference was larger (16.5% of annual SWI occurred after the stream dried), which is partly because the dry-out date of the stream was ~1.5 month earlier than in the other years (13 July vs. 25-26 August).

We adapted the calculation of SWI in the revised manuscript and updated the corresponding text on (L362-364): "The dry-out date of the stream was significantly correlated to [...] the sum of SWI before the dry-out date" and in the explanation of the metrics (L264-265): "To compare SWI with the dry-out date, we also calculated how much SWI occurred during the water year before the stream dried."

**1.7 Figures that better illustrate main findings** Although, I found the reasoning presented in results and discussion sections correct, the supporting illustrations are, in my opinion, less informative and I am not sure whether they fully support all the results and interpretation. For example, one of the main conclusions is that temporal distribution of SWI is more important than its total amount. While I agree with that, it is

difficult to me to clearly see this in figures which mostly shows only time series (Figs. 4 and Fig.5). I do not have any clear suggestion how to make figures more informative and supporting the results, but I would encourage authors to reconsider their illustrations and perhaps add another figure which would better show how the timing of SWI influence the runoff response.

Thank you for this remark. It is for us very important that we capture our main findings in the figures and appreciate your comment to make us aware of this. We added a heatplot (Fig R1.2 included in main text as Fig. 7, pasted below) to better match the main findings with the figures. We think that it covers the relevant correlations and metrics relating to the temporal distribution of SWI, as well as the annual metrics for SWI and snowfall.

[Figure]

Fig. 7: Heatplot showing Pearson correlation coefficients (α=0.1) for comparisons between annual discharge, the stream dry-out date and precipitation and snowpack metrics. Significant correlations are marked in dark red (negative) and dark blue (positive), whereas insignificant correlations are marked in light blue (positive) or light red (negative) and correlations without a direction are marked in white (r < 0.3). For most metrics, the comparison is based on the 2004 2014 data record (n=11 years). The comparison with the melt-out date (marked with one asterisk) is based on the simulated years (n=5) and the years for which satellite imagery was available (2016-2019, n=4; which totals to n=9). For the SWI flashiness index, the melt rate, and the number of days when at least half the catchment was snow-covered and the sum of SWI before the dry-out date (marked with two asterisks), we used only the years that were simulated (n=5). Scatter plots of all significant correlations can be found in Supplemental Fig. S9.

**Technical corrections**

L 116: The decrease in streamflow should be expressed in mm/decade to be comparable with other characteristics.

Thank you for pointing this out. We now divided the trend (-0.75 * $10^6$ $m^3$/decade) by the surface area (54.44 $km^2$) and now include the trend as -13.8 mm/decade in the text of the revised manuscript (L138).

L 138: "stage height-discharge relationship". Maybe more common term "rating curve" would be better.

We don't hold any preference between the two terms since we are familiar with both, but adapted it to rating curve in the revised manuscript to avoid confusion with any future readers (L163).

L 193: "Trujillo et al. (2019, manuscript in preparation)". As it seems from references, this paper has been already published.

The Trujillo et al., (2019) reference refers to an AGU abstract, which has been presented at the 2019 AGU Fall Meeting. We added "AGU Fall Meeting" to the bibliography so that this is clear. The corresponding manuscript is still in preparation, and we removed it from the reference list.

Fig. 6a: The annual discharge is related to the precipitation at jdt125 climate station. Why not to show catchment mean precipitation instead? If I understood correctly, the model interpolates stational data to a catchment scale using some kind of elevation dependency. Therefore, to show catchment precipitation in Fig. 6a makes more sense to me to make it better comparable to catchment runoff.

We used precipitation at the climate station rather than simulated precipitation so that we could include additional years in the dataset (2004-2014) without having to run the model for the additional years. Catchment-average precipitation for the years that we did model was linearly related ($R^2$: 0.93) to the precipitation at jdt125 (Figure R1.4, now included as Supplement S11). That precipitation at this lowest elevation station is slightly lower than the simulated catchment-average precipitation, based on four stations, is not surprising, since precipitation increases with elevation. All in all, the strong correlation indicates that using precipitation at this station is not expected to lead to a different interpretation. We refer to supplement S11 in the caption of Fig. 6 in the revised manuscript.

[Figure]

**Figure S11 in the revised supplement**: Precipitation at jdt125 (the low elevation precipitation gauge) versus the simulated mean catchment precipitation for the years that were modeled.

Fig. 6b: What the triangles represent? Maybe, there is a mistake in the figure as they represent "other years", but different symbol is used in the legend.

This was a mistake in the legend. The diamonds in the legend (other years, 2016-2019) should have been reversed triangles. We corrected this mistake and checked the legends and symbols in all figures before resubmitting.

**References**

Baker, D. B., Richards, R. P., Loftus, T. T., and Kramer, J. W.: A NEW FLASHINESS INDEX: CHARACTERISTICS AND APPLICATIONS TO MIDWESTERN RIVERS AND STREAMS, J Am Water Resources Assoc, 40, 503–522, https://doi.org/10.1111/j.1752-1688.2004.tb01046.x, 2004.

Berghuijs, W. R., Woods, R. A., and Hrachowitz, M.: A precipitation shift from snow towards rain leads to a decrease in streamflow, Nature Clim. Change, 4, 583–586, https://doi.org/10.1038/nclimate2246, 2014.

Godsey, S. E., Marks, D., Kormos, P. R., Seyfried, M. S., Enslin, C. L., Winstral, A. H., McNamara, J. P., and Link, T. E.: Eleven years of mountain weather, snow, soil moisture and streamflow data from the rain–snow transition zone – the Johnston Draw catchment, Reynolds Creek Experimental Watershed and Critical Zone Observatory, USA, Earth Syst. Sci. Data, 10, 2018.

Irannezhad, M., Marttila, H., and Kløve, B.: Long-term variations and trends in precipitation in Finland, Int. J. Climatol., 34, 3139–3153, https://doi.org/10.1002/joc.3902, 2014.

Trujillo, E. and Molotch, N. P.: Snowpack regimes of the Western United States, Water Resour. Res., 50, 5611–5623, https://doi.org/10.1002/2013WR014753, 2014.

**Reviewer 2**

Dear reviewer,

Thank you for taking the time to review our paper and for your constructive comments. Please find our responses (in blue) to your specific comments (in black) below. The line numbers detailing where to find adaptations in the main text of the revised manuscript are printed in red.

Kind regards on behalf of all co-authors,

Leonie Kiewiet

**2.1 Main finding and figures***:* I agree with reviewer #1 that the text at some places in the results and discussion sections read clear and sound, but the figures do not always support the conclusions or findings. As mentioned by reviewer 1 especially the conclusion on the temporal distribution of SWI is not easy to extract from the figures. Figure 4 and 5 show SWI and discharge, but because the years differ in so many aspects (ratio of rain and snowmelt, timing of SWI, variability of SWI, Q), it is hard to tell which process caused the discharge response from the timeseries, i.e. to see a clear link between temporal distribution and discharge. The text describes these different aspects, but how to generalize these results more? Maybe some measures related to the timing of the center of volume for rainfall and snowmelt, antecedent conditions before spring or number and timing of melt/rainfall events could give some insights. Could also some measure on spatial and temporal distribution be combined? Probably the authors know best how they drew this particular conclusion and could use that to focus on that aspect in the results/figures more explicitly.

We attempted a clearer presentation of the findings by revising the two existing figures and adding one new figure. We revised Figure 4 to add an indication of the timing of the center of mass for SWI, rain, and discharge. We also added a new heatplot to the revised manuscript (Fig. 7, pasted above in reply to review comment 1.7), based on your comment and the comment of reviewer 1. We include the fraction of annual SWI occurring in each season as an additional metric, as well as snowfall in spring and winter and the melt rate (see reply to review comment 1.7). In this new figure, we evaluate how different metrics of the (temporal) distribution of SWI are linked to both annual stream discharge and the day that the stream dried up. This new figure shows how different aspects of the temporal distribution of SWI connect to discharge or dry-out date because the figure highlights statistically significant and insignificant correlations.

**2.2 Study setup***:* While going through the manuscript I was wondering why only four years were selected. Because of the many processes that influence the discharge signal, a larger sample of years may have provided stronger evidence how processes relate, i.e. avoid that for example the dry year that was analyzed had many rainfall events. From the data description it is a bit unclear to me what the maximum possible amount of years could have been for analyzing. The decision may have to do with the runtime of the model? At least I would expect some description how the selected years deviate from the mean hydro-climatology of the catchment. Maybe the discussion/limitations section could elaborate on the selection of the years and the intertwined processes when looking at observations and possibilities for future model experiments, isolating some of these aspects (for which discharge would needed to be simulated as well) – but this last point as the authors see fit.

We selected four years because setting up and running the model was a non-trivial task. Also, we aimed to focus on differences in the distribution of SWI and stream discharge for years that had different snowfall ratios and total water inputs and therefore, we selected strongly contrasting years from the 11 potential years of record (Godsey et al., 2018). We now describe this rationale in the revised manuscript (L215-220). A summary of how the selected years differed from the long-term average is now briefly summarized in the methods section of the revised manuscript (L212-215) and elaborated in Supplemental Table S3 (pasted below), which is referred to in the revised main manuscript on L214. We also now include a scatter plot of annual snow fraction and annual precipitation that shows that the years we chose to simulate contrasted with the other years captured in the dataset in the supplementary material (Figure S4, pasted below).

Although 2007 was slightly drier than 2014 and 2006 was slightly wetter than 2011, we chose to simulate 2011 and 2014 because additional weather stations had been installed in 2011. Temperature and humidity data from these additional stations increased model accuracy, and snow depth data from these locations was used to validate the model outputs.

**Table S3** in the revised supplement: Annual precipitation (P, mm), snowfall fractions (SF, -) and air temperature ($T_a$, °C), as well as % of the mean of the 2004-2014 record (Godsey et al., 2018). Simulated years are printed bold.

| | P (mm) | % P | SF (-) | % SF | $T_a$ (°C) | % $T_a$ |
|---|---|---|---|---|---|---|
| 2004-2014 | 524 | 100 | 0.37 | 100 | 8.2 | 100 |
| 2004 | 470 | 90 | 0.49 | 132 | 8.4 | 103 |
| 2005 | 543 | 104 | 0.23 | 63 | 8.2 | 100 |
| 2006 | 714 | 136 | 0.29 | 78 | 8.4 | 103 |
| 2007 | 402 | 77 | 0.31 | 83 | 9.3 | 113 |
| 2008 | 465 | 89 | 0.45 | 123 | 7.4 | 91 |
| 2009 | 549 | 105 | 0.49 | 132 | 8.0 | 98 |
| 2010 | 531 | 101 | 0.57 | 155 | 6.6 | 81 |
| 2011 | 693 | 132 | 0.41 | 111 | 7.4 | 91 |
| 2012 | 494 | 94 | 0.24 | 64 | 8.6 | 105 |
| 2013 | 456 | 87 | 0.26 | 72 | 8.6 | 105 |
| 2014 | 450 | 86 | 0.30 | 82 | 8.6 | 105 |

[Figure]

**Figure S4**: Scatterplot of the annual precipitation and snowfall fraction of precipitation at weather station jd125, which is located close to the catchment outlet. Simulated years are shown in blue, other years are shown in black.

**2.3 Argumentation in introduction***: Partly related to the comment of reviewer 1 on a better description of the novelty in the introduction, I think that the line of thoughts for this study and the research gap can be better described. In my opinion, the introduction mixes 1) changes in snowmelt generated streamflow, 2) differences between catchments seasonally snow covered and in the rain-to-snow transition zone, 3) rain-to-snow zones as a space-for-time substitution of catchments that are now seasonally snow covered and 4) changes that have occurred in the rain-to-snow transition zone and may occur in the future. Although all of these aspects may be important to put the study into context, I would suggest to clearly identify the research gap (how do yearly variations in rainfall and snowmelt influence discharge, relation with snowfall fraction not yet clear, rain-to-snow zone suitable to analyze 'extremes', i.e. snowy and rainy) and explain the implications for future changes and relations to observed changes in different type of catchments in a more structured way.

Thank you for this comment. We agree that it is important to clearly describe the research gaps and rationale behind the study, and restructured the introduction to follow these steps:

1. Climate change affects snowfall fractions, but it remains unclear how sensitive stream discharge is to these changes.
2. Stream discharge might be affected by changes in the temporal distribution of water inputs, because it affects water partitioning (e.g., precipitation goes to streamflow, evaporation or groundwater recharge).
3. Stream discharge might be affected by changes in the spatial distribution of water inputs, and in particular snow drifts, that have been shown to affect streamflow in the summer season and control local groundwater levels.
4. The rain-snow transition zone is a place where the effect of variations in snowfall fractions can be studied effectively, since it experiences large year-to-year variations in snowfall fractions.

After these steps, we introduce the research questions, and follow with a short statement that emphasizes the novelty of the work (L123-131).

**2.4 Methods and data description***: Here I missed some details regarding the available data, the model and the choice of years. As indicated above, it is not mentioned how the four climatologically different years were selected. I was also a bit confused by the numbers in table 1, how come that in a rainy year, the SWI_snow is higher than in a snowy year? Are numbers switched here? And without knowing the range of snowfall fractions over a longer time period it is difficult to interpret the values of the different years. It would also be helpful to explain the reasoning and possible hypotheses of selecting rainy and snowy years and wet and dry years. Could temperatures also be given for the years? Regarding the data and model,

what is needed as input for the model? And which of the stations do have this data available for which time period.

We recognize that the description of the data, model and selection of years was rather short. We added more information on the functioning of the model in (L235-251, described above at comment 1.2), as well as information on the required model input and how that overlaps with availability of data for the different years (L218-220). We also expanded the description of the selection of the simulated years, as explained above at comment 2.2 (L210-221). This should provide context for the years that we chose to simulate as well as on the dataset as a whole.

Thank you for pointing out the mistake in Table 1. It should have been 412 mm and 243 mm for SWI from rain in 2005 and 2010, respectively, and 145 mm and 310 mm for SWI from snowmelt in 2005 and 2010, respectively.

*Minor and technical corrections:*

**2.5 Title + abstract: 'Snowfall fractions'** – since you only clarify in the introduction, maybe another term could be used here, e.g. ratio of snowfall to precipitation. Regarding the title, maybe it needs to be adjusted depending on the changes, e.g. temporal distribution and total input? Or specify what is meant with temporal distribution. Stream discharge – Annual (stream) discharge.

Although some recent articles do use the term 'snowfall fractions' in their titles (e.g., Nolin et al., 2019, Jennings et al., 2020), we recognize how this might be confusing, and now removed this term from the title and abstract. We also described discharge more explicitly so that the new title reads:

"Effects of spatial and temporal variability in surface water inputs on streamflow generation and cessation in the rain-snow transition zone"

L13 '..spatial and temporal distribution of precipitation' – add phase of precipitation?

Yes indeed, precipitation phase might also impact stream discharge. We rewrote this part of the abstract so that the first sentence now reads (L12):

"Climate change affects precipitation phase, which can propagate into changes in streamflow timing and magnitude."

L68 which catchments?

We think that our findings will be most applicable to other small (<10 km$^2$), semi-arid, mid-elevation, mid-latitude catchments, and will include that as specification. We suggest these catchment characteristics because we think that 1. similarly sized catchments are more likely to have a similar potential for water storage on the

surface and in the subsurface, 2. semi-arid, mid-latitude catchments are likely to have a similar vegetation cover 3. mid-latitude, mid-elevation catchments are likely to have a temporal distribution of water inputs that is similar to that in Johnston Draw. This is now detailed on L110: "…thereby provide insight in how other small (<10 km$^2$) catchments with a similar vegetation cover and precipitation regime might respond to future changes in rain/snow apportionments"

L71-72: on an annual time-scale is this so different, apart from the effects of snow redistribution? Is this something interesting to show for you analyses, i.e. spatial distribution of rainfall and spatial distribution of snowmelt?

Apart from the snow redistribution, distribution of rainfall and snowfall might be quite similar across the catchment, and in both cases increase slightly with elevation. However, snow redistribution causes surface water inputs to differ across the catchment, even at the rain-snow transition zone. We added the following sentence to make this clear to the reader:

L65-66: "Hence, differences in the SWI distribution due to varying snow depths could be particularly substantial in areas where wind-driven redistribution of snowfall is significant."

When analyzing the data, we visualized SWI inputs from rainfall and snowmelt separately, but refrained from including these figures in the manuscript. We decided this because rainfall amounts were interpolated following a linear orographic gradient derived from precipitation at the gauges at the upper and lower end of the catchment. Because of this, rainfall distribution across the catchment did not reflect any small-scale variations in the spatial distribution of rainfall, other than that caused by elevation. The effects of wind-redistribution of snow were implicitly included in the spatial distribution of snowfall, which was based on the lidar snow depths, and thus included more fine-scale spatial variation. Hence, a direct comparison with the orographic precipitation gradient might overrepresent the differences. Investigating how the spatial distribution of non-redistributed snowfall and rainfall might differ could be achieved by simulating the snowpack with and without wind re-distribution, but we think this is outside of the scope of the manuscript presented here.

L94: 'However' – where does this refer to?

This was meant to refer to the difference in catchment wetness that might exist between rainfall versus snowmelt dominated catchments. We removed the word 'however' so that the sentence now reads: "Rain and snowmelt inputs might result in similar runoff ratios (discharge/SWI) as long as the overall catchment wetness is similar or if the catchment is wet at key locations for water transport." (L51-53)

L116-117: did increased ET played a role here?

While the long-term analysis of Nayak et al. (2010) does not comment on increased evaporation or transpiration, Seyfried et al. (2011) states that evapotranspiration is most sensitive to increases in PET (implied by increases in air temperature) during ~4-5 weeks each year in which the plants have developed leaves and sufficient water is available in the soil. Before that time, plants use little water, and after that time, the system is strongly water-limited. Hence, although increased plant water use might be important in some systems, we suspect that it might not strongly affect stream discharge in this region. We summarized this information on L139-141 of the revised manuscript: "These streamflow trends are unlikely to be driven by increased plant water use (caused by increased temperatures) because there is only a short time window (~weeks) in which plant leaf-out has occurred and there is still sufficient soil water available in this water-limited environment (Seyfried et al., 2011)."

L195-196: 'this uncertainty.... Patterns' – double with few sentences above

This part of the sentence repeats itself because we wanted to highlight the connection between the intra-annual consistency in snowpack patterns introduced above, and the uncertainty related to using the snow-on lidar from only one year, discussed here. To avoid the exact repetition of words, we changed this part of the sentence (L230-233) to read: "Although using the 2009 survey to rescale snowfall in other years might have induced some uncertainty, verification of the interannual consistency in the snow distribution in this catchment by comparing the lidar snow depth and the satellite imagery indicated that this uncertainty is likely to be small.."

Section 3.5 How do catchment precipitation and discharge compare? Are there estimations for ET?

Precipitation and discharge are significantly correlated ($R^2$= 0.6, p-value = 0.005), which is shown in Fig. 6a of the original manuscript (and retained in revision). There are no estimations for ET in this catchment. However, there are estimations for a nearby catchment at slightly higher elevation (1930 m) and that receives less precipitation (Upper Sheep Creek; Flerchinger and Cooley, 2000; MAP: 479 mm versus 609 mm for Johnston Draw). They showed that evapotranspiration accounts for nearly 90% of effective precipitation. Although estimating ET for the years and catchment presented here is beyond the scope of this effort, we included this information about Upper Sheep Creek in the discussion of the revised manuscript (L355-356).

L223 'this pattern was masked by the effects of other processes' – what is meant here? In general in the results section it would be helpful to indicate better when observations or when simulations are described.

Here we mean the snow redistribution processes, and we further detailed measurements in which this process can be observed in L223-228 of the initial

manuscript. We agree that being specific is helpful, and we changed the text to "...the snowpack distribution was also affected by wind-driven redistribution of snow. For instance, the snow depths at jdt2 ...." (L281-282)

L236-237 'differential melt-out patterns' – what was compared for that?

We compared the simulated persistence of the snowpack with the persistence of the snow-covered area from the satellite imagery. This comparison showed that the areas that were simulated to be snow-covered longer were also snow-covered longer in the satellite imagery. We recognize that we did not explain that very clearly in the manuscript and added the following text to section 4.2 (L291-297) of the revised manuscript:

"The spatial pattern of the lidar snow depth also agreed well with the spatial patterns of snow-covered area (Fig. 2a,d), and there was a strong agreement between the simulated snow-covered area for 2009 (Fig. 2e) and the snow-covered area determined from satellite imagery for 2019 (Fig. 2d), including the modelled duration of snow cover and the number of satellite images in which snow-covered areas were observed. The largest discrepancy between the simulated and imagery-based snow duration was in the scour zone west of the snow drifts, where the model underestimated snow duration. Nonetheless, the consistent locations of the snow drifts between 2009 and 2019 indicates that the model captured the spatial distribution of the snowpack."

L267 'As a result, average daily SWI rates were higher' – as a result of what?

We meant to say here that average daily SWI rates were higher as a result of higher snowfall and lower rainfall inputs, and adapted this sentence so that it reads: "Average daily SWI rates were higher in snowy 2010 than in rainy 2005 (mean SWI rate March-May: 3.7 mm d$^{-1}$ in 2010 vs. 2.9 mm d$^{-1}$ in 2005)." (L330-331)

L274 'whereas roughly 30% of SWI....' – are delays taken into account, or is meant here the comparison between SWI from month x to month y and discharge from month x to month y? Are the events where Q is higher than SWI also of interest?

This refers to the comparison of SWI in period x and discharge in period x; no delays are taken into account. We substantially revised this section and removed the original language; this comparison is now outlined in the methods (L261-271):

"The phase and magnitude of precipitation and the magnitude and temporal distribution of SWI were compared to annual discharge and the stream dry-out date. The stream dry-out date is the day when the stream first ceased to flow at the catchment outlet. For comparisons across seasons, we defined winter as December, January and February; spring as March, April and May; summer as June, July and August, and fall as September, October and November. To compare SWI with the

dry-out date, we also calculated how much SWI occurred during the water year before the stream dried. No delays were considered when comparing SWI to discharge (e.g., discharge as a fraction of SWI in January results from dividing discharge in January by SWI in January). Discharge metrics were also compared to the flashiness of SWI inputs, which was calculated as the sum of the difference in total SWI from day to day, divided by the sum of SWI (also known as the Richards-Baker Flashiness Index; Baker et al., 2004). Further metrics included the fraction of time that more than half of the catchment was snow-covered and the melt-rate between 40% snow-coverage in the catchment and the date at which the catchment was snow-free. A threshold of 40% snow-coverage was chosen because this resulted in an approximately linear melt-rate for all years."

Although events where Q is higher than SWI are definitely of interest for future work, they are not further explored in this manuscript because we did not investigate discharge generation during individual events.

L279 Have you tried plotting % of SWI translated into discharge against temperature (annual, or during growing season?)

We did not do this, but find it an interesting suggestion! Plotting runoff efficiency as discharge/precipitation vs. mean air temperature shows that they are weakly and insignificantly correlated ($R^2$= -0.43, p-value=0.217; Fig. S11, pasted below). Perhaps this corroborates that evapotranspiration is water-limited in this system rather than energy-limited. We also investigated if the runoff ratio was related to precipitation or SWI on the seasonal scale (i.e., spring, summer, fall winter, Fig. S10 in the revised supplement). We found that Q/P was weakly correlated to air temperatures during each season, with the summer period yielding the highest correlation ($R^2$=-0.54, p-value=0.08). We assume that temporal offsets between snowfall and snowmelt might have led to low correlations in spring and winter. We found strong but insignificant relationships when calculating the efficiency as Q/SWI (up to $R^2$=-0.72, p-value=0.169, also during the summer season), and suspect that the insignificance of this relationship is likely due to the low number of observations (n=5). Together, these results suggest that temperature likely influences runoff efficiency in the warmer season by increasing evapotranspiration, and might affect winter runoff efficiencies by causing faster snowmelt.

We included Figure S11 in the revised supplemental materials and alluded to this small additional analysis in the discussion of the revised manuscript (L415-422):

"The runoff efficiency, calculated as summer discharge divided by summer precipitation for the 2004-2014 record, was significantly correlated to summer air temperatures (r2=-0.54, p value=0.08, Supplemental Fig. S10) whereas this relationship was insignificant on the annual scale (r2= -0.43, p-value=0.217; Supplemental Fig. S11). This suggests that evapotranspiration, which is directly

affected by the ambient air temperature, has some influence on runoff efficiency, despite the catchment being an overall water-limited environment. In winter, higher temperatures result in higher runoff efficiencies (r2=0.48, p-value=0.131, Supplemental Fig. S10), which is likely due to faster melt-out and more saturated soils, as described above. However, further simulations are required to fully understand how precipitation amounts, timing and location interact with subsurface water storage to control stream discharge."

[Figure]

**Figure S11 in the revised supplement:** Annual air temperature (°C) versus runoff efficiency (discharge/precipitation, mm mm$^{-1}$).

[Figure]

**Figure S10 in the revised supplement:** Seasonal air temperatures (°C) versus runoff efficiencies (discharge/precipitation, mm mm$^{-1}$).

Section 5 – the subsections have no numbering

Thank you. We added numbers to the subsections in the discussion.

L357 'This highlights the importance of the temporal distribution of SWI' – also the importance of total water input?

Definitely! Although this was not emphasized in the initial version of the manuscript, we now added this to the results (L445-446) and the conclusions (L515-516).

L360 'events' – throughout the manuscript when using 'event' please check if it is clear why event is meant? Precipitation, rainfall, snowmelt, discharge?

Thanks for pointing this out. We meant 'event' as 'rainfall or snowmelt event' (i.e., an event related to SWI). We now specify the type of event at each of the eight occurrences in the manuscript (e.g., it is written explicitly in the text as precipitation event, rain-on-snow event, snowmelt event...).

L369 'catchment' – sub-catchment?

Yes, 'Treeline' refers to a sub-catchment rather than the entire Dry Creek catchment. We updated this in the text of the revised manuscript (L441).

Discussion on simulated snow depts – could it be extended with a description of the reasons for varying performance for individual years and maybe a hypothesis how such 'bad' simulated years potentially could have influenced the results?

Thank you for this suggestion. We think that a summary of the extensive discussion in our reply to reviewer 1 is valuable addition to the current discussion on simulated snow depths, and have included this in L450-461 of the revised manuscript as noted above in Response to Reviews section 1.3.

The effect of the varying performance could also vary between years. For instance, in years in which the actual snow redistribution was less strong than simulated, snow drifts might have been overrepresented, resulting in a later simulated melt-out date of the snowpack. If the snow redistribution was overestimated in some years, but underestimated in others, this might have resulted in either a stronger or weaker relationship between the snowpack melt-out date and the stream dry-out date.

L419 '..., which influences' – should it be, which may influence? As for example one of your conclusions is that the spatial distribution of SWI stays rather stable over time?

Thank you for this suggestion. Indeed, although we might expect that effects on the spatial distribution could be more severe if snow redistribution patterns are also affected, that is not shown in our work. We adapted the phrase to "may influence" (L505).

L428 -429 Could a short explanation/hypothesis be added why Q was much higher in 2010?

We have substantially changed the conclusions so that this comment does not apply to the conclusions anymore.

We now do explain why discharge in 2010 might have been very high, in L404-408 of the revised manuscript "The steadier water inputs in the snowmelt period might explain why annual discharge in snowy 2010 was double that of rainy 2005 despite similar total SWI. More stable water inputs from snowmelt rather than flashy water inputs from rain could have led to wetter soils and higher soil conductivity rates, allowing more water to pass through the subsurface towards the stream or towards deeper storage (Hammond et al., 2019)."

We also checked if the fraction of precipitation occurring in spring was related to annual stream discharge or runoff efficiency, and found a statistically significant positive relationship with the stream dry-out date ($R^2$= 0.58 p-value=0.06), and a positive, statistically insignificant relationship for annual discharge; $R^2$=0.43 p-value=0.18). We now include these findings in the heatplot shown in Fig. 7 and the scatter plots in Supplemental Fig. S9.

Figure 2e – what do the light coloured pixels mean? Was there no snow cover in the simulations while there was around 0.5 in the satellite observations? Because of the comparison of different years?

The light-colored pixels indicate that the time that an area was simulated to be snow-covered was lower than 0.25 (0-0.25). Indeed, that estimation differs from the fraction of time that these areas were snow-covered based on the satellite imagery (0.5), and we now highlight this difference in L296-296. This difference might be due to a small difference in the mean annual air temperature, which was a bit higher in 2009 than in 2019 (8.0°C versus 6.7°C), which could have resulted in faster melt-out. We highlight this in L180-282 of the revised manuscript. "This targeted year was warmer than the year for which the lidar observations were available (mean annual air temperature: 8.0°C compared to 6.7°C in 2009), which may have resulted in earlier peak streamflow, melt-out date, and dry-out date for the stream."

For all figures it may be good to not only indicate the year but also its characteristic (i.e. snowy, rainy, wet and dry) in the figure itself instead of the legend.

Thank you for this suggestion. In earlier versions of the figures we indeed included the characteristic of each year in the panel titles, and re-introduced that for the figures in the revised main manuscript.

---

## Author Response (AR2)

Dear Markus,

Please find with this upload the final version of the manuscript, in which we addressed the points that were raised by reviewer 1. We refrained from changing the figure captions, since we think that extra clarity on the meaning of different symbols and colors (although captured in the legends as well) could help future readers to orient themselves more efficiently.

Many thanks again for your efforts in handling our manuscript.

On behalf of all co-authors,

Leonie Kiewiet